# RETRIEVAL-OF-THOUGHT: EFFICIENT REASONING VIA REUSING THOUGHTS

**Ammar Ahmed[1,*], Azal Ahmad Khan[1,*], Ayaan Ahmad[2], Sheng Di[3], Zirui Liu[1], Ali Anwar[1]**

[1]Department of Computer Science and Engineering, University of Minnesota

[2]Department of Computer Science and Engineering, University of California Santa Cruz

[3]Argonne National Laboratory

[*] Equal Contribution.

{ahme0599, khan1069, zrliu, aanwar}@umn.edu
ayahmad@ucsc.edu, sdi1@anl.gov

## ABSTRACT

Large reasoning models improve accuracy by producing long reasoning traces, but this inflates latency and cost, motivating inference-time efficiency. We propose Retrieval-of-Thought (RoT), which reuses prior reasoning as composable "thought" steps to guide new problems. RoT organizes steps into a thought graph with sequential and semantic edges to enable fast retrieval and flexible recombination. At inference, RoT retrieves query-relevant nodes and applies reward-guided traversal to assemble a problem-specific template that guides generation. This dynamic template reuse reduces redundant exploration and, therefore, reduces output tokens while preserving accuracy. We evaluate RoT on reasoning benchmarks with multiple models, measuring accuracy, token usage, latency, and memory overhead. Findings show small prompt growth but substantial efficiency gains, with RoT reducing output tokens by up to 40%, inference latency by 82%, and cost by 59% while maintaining accuracy. RoT establishes a scalable paradigm for efficient LRM reasoning via dynamic template construction through retrieval. The code for RoT is available at https://github.com/ahme0599/Retrieval-of-Thought

## 1 INTRODUCTION

Large Reasoning Models (LRMs) have demonstrated impressive capabilities in solving complex tasks by producing outputs accompanied by detailed reasoning trajectories (Xu et al., 2025a). Proprietary models such as OpenAI's o1, o3, and o4 (Jaech et al., 2024), Google's Gemini 2.5 (Comanici et al., 2025), and Anthropic's Claude Opus 4, as well as open-source models like Qwen-QwQ (Yang et al., 2025a) and DeepSeek-R1 (Guo et al., 2025), exemplify this trend. These models adopt an intentionally slower and more deliberative inference process, mimicking human-like reasoning. This approach typically involves generating longer outputs and consuming increased inference-time compute to effectively address reasoning-intensive queries.

Recent efforts to improve reasoning in LLMs have primarily focused on generating more output tokens to simulate thoughtful, multi-step reasoning (Snell et al., 2024). A common approach involves guiding generation using external reward models Zhang et al. (2024). These include outcome-based reward models, such as Best-of-N (BoN) sampling. Other methods employ process-based reward models that supervise intermediate reasoning steps through search strategies like beam search or Monte Carlo Tree Search (MCTS) Xie et al. (2024); Zhang et al. (2025b). More recently, reinforcement learning (RL)-based techniques, such as Group Relative Policy Optimization (GRPO), have been explored to train models toward trajectories that yield high reasoning quality (Shao et al., 2024). While increasing test-time compute has improved reasoning performance, this often relies on generating longer outputs, which introduces key inefficiencies (Sui et al., 2025; Feng et al., 2025). First, generating longer outputs inherently increases latency, as each token must be decoded sequentially. Second, API-based model providers typically price output tokens 2–5× higher than input tokens, making such methods cost-prohibitive at scale. These limitations highlight the need for more efficient reasoning techniques that retain performance while reducing generation cost and latency.

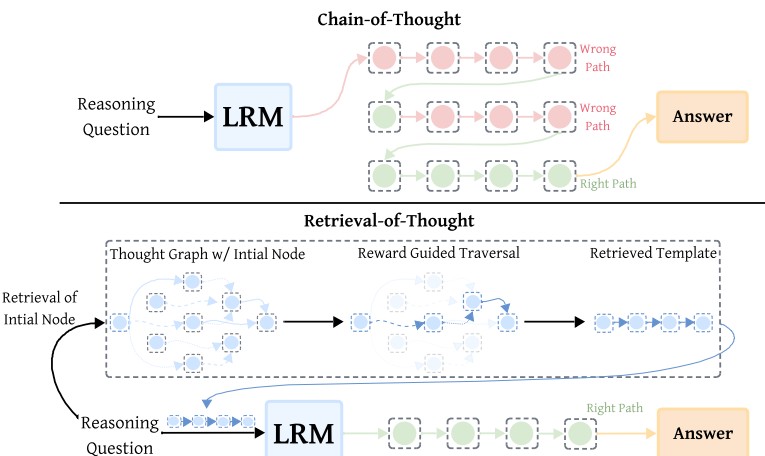

Figure 1: **The figure contrasts Chain-of-Thought (CoT) inference in LRMs with our Retrieval-of-Thought (RoT) approach.** In CoT **(top)**, models sequentially explore multiple wrong paths, causing inefficiency and high token usage. RoT **(bottom)** builds on a structured thought graph where reasoning steps are stored as nodes. First, RoT retrieves relevant nodes and performs reward-guided traversal to assemble a problem-specific template, reducing redundant exploration and directing the model toward correct reasoning. This yields more efficient inference and fewer tokens.

To enable more efficient reasoning without excessive token generation, a promising direction is to reuse prior knowledge in the form of thought templates. Recent work has pursued this via retrieval-based reasoning, Buffer of Thought (BoT; Yang et al. (2024a)) and SuperCorrect (Yang et al., 2024b) augments this with both coarse and fine-grained templates to bolster smaller LLMs. However, these approaches operate largely with static templates fixed before generation, limiting their ability to adapt or compose new reasoning pathways on the fly. In contrast, human problem solving relies not only on recalling past experiences but also on "connecting the dots", recombining fragments of prior solutions into novel configurations that make knowledge broadly reusable (Gick & Holyoak, 1980). Translating this to LLMs, existing methods lack the ability to dynamically assemble templates across problems, leaving them unable to exploit the kind of flexible memory that humans use to reason efficiently. To address this gap, we propose RoT, a framework that equips LRMs with a structured memory of individual thoughts and enables the dynamic construction of new templates through a knowledge graph, thereby supporting more efficient and generalizable reasoning.

The foundation of RoT is built upon three key observations related to performance optimization. **(O1)** Queries within a given dataset often exhibit recurring patterns in the underlying reasoning or thoughts[1] used to solve them. **(O2)** Retrieving relevant information from a large vector database is significantly faster than generating text from a LM. **(O3)** When presented with a correct reasoning path or solution to a similar query, language models can solve new queries more efficiently, requiring fewer tokens to reach a correct or complete answer. Yet fully exploiting these observations exposes a core challenge to design a system that can dynamically generate reasoning templates during inference by observing the input, rather than reusing fixed scaffolds. This requires not only storing a rich repository of individual reasoning thoughts but also developing mechanisms to identify, retrieve, and synthesize the most relevant patterns into coherent, query-specific templates. Unlike prior approaches, RoT aims to provide LRMs with the ability to construct contextually appropriate reasoning scaffolds on-demand by flexibly combining and adapting prior thoughts, thereby enabling efficient, reusable, and generalizable reasoning.

As shown in Figure 1, we introduce RoT's novel architecture, which organizes reasoning steps into a thought graph. This graph leverages metadata-based retrieval to quickly narrow the search space and then applies reward-guided traversal algorithms to dynamically assemble thought templates by exploring interconnected reasoning patterns. The resulting templates are generated at inference

---

[1]Throughout this paper, we use the term *thought* interchangeably with *reasoning step*, referring to the individual steps used to solve a reasoning problem.

time, ensuring both contextual relevance to the query and computational efficiency. We further demonstrate how these templates can be applied to guide reasoning, reducing token generation and establishing RoT as a more efficient framework for LRM problem solving.

In CoT method LRMs frequently explore through multiple incorrect reasoning paths before converging, whereas RoT uses the retrieved template as a guide that anchors the model to promising reasoning paths. This substantially reduces path switching, enabling the model to reach the correct solution with fewer detours. As a result, RoT lowers the number of output tokens generated, directly translating to improved efficiency in both cost and latency.

Our work makes three key contributions to efficient inference in reasoning models. ***First***, we propose a new paradigm for dynamic template construction, introducing the first framework that equips LRMs with a structured memory of reasoning and dynamically assembles context-specific templates at inference time, unlike prior methods that rely on static templates. ***Second***, we design a thought graph with a reward-guided traversal algorithm that organizes reasoning traces and efficiently retrieves and composes templates while preserving contextual relevance. ***Finally***, through comprehensive experiments and ablations across AIME and AMC benchmarks, we show that RoT consistently achieves comparable or higher accuracy than Chain-of-Thought and retrieval baselines, while reducing output tokens by up to 20–40% and cutting inference latency and cost substantially. These results establish RoT as a scalable and efficient solution for LRM reasoning.

## 2 RELATED WORKS

**Test-Time Scaling.** Scaling test-time compute has emerged as a key strategy for improving reasoning, with methods such as Best-of-N sampling, beam search, and more recently RL approaches like GRPO enabling models to generate longer reasoning traces and improve accuracy (Snell et al., 2024; Yao et al., 2023; Guo et al., 2025). While effective, these methods come at the expense of efficiency as longer outputs substantially increases latency and cost, motivating recent work on inference-time efficiency through adaptive computation, parallel decoding, and early termination strategies (Pan et al., 2025; Chen et al., 2025; Fang et al., 2025). Together, these efforts highlight a growing demand for techniques that preserve reasoning performance while reducing computation.

**Retrieval-based Reasoning.** Retrieval-based methods represent a complementary direction, injecting external knowledge or pre-computed reasoning to guide inference. Prior approaches such as BoT (Yang et al., 2024a), SuperCorrect (Yang et al., 2024b), and RAT (Wang et al., 2024) have shown that reusing thought templates can improve reasoning, though they typically rely on static scaffolds distilled from larger teacher models. In contrast, our work targets reasoning-capable LMs directly and introduces a dynamic alternative: decomposing prior solutions into reusable thought steps and assembling problem-specific templates at inference time. This fine-grained, on-demand reuse enables greater adaptability than static retrieval, establishing a new paradigm for efficient reasoning. A complete discussion of related work is provided in Appendix A.

## 3 MOTIVATION

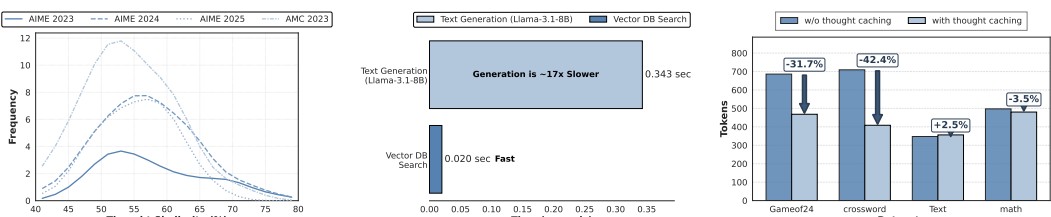

Figure 2: Key observations motivating the RoT framework. **(Left)** Semantic similarity between Thought Store and steps to solve reasoning datasets. **(Middle)** Comparison of retrieval and generation showing retrieval is faster than generation. **(Right)** Thought Caching yields token savings.

Our framework builds on three central observations about reasoning efficiency.

**(O1) Thoughts required to solve problems across same domain are highly similar.** Problems in reasoning datasets often require overlapping steps. For instance, many AIME or AMC questions share algebraic transformations or simplification patterns. When we analyze solution traces, we see that the same types of steps appear again and again, as shown in Figure 2 (left). This suggests that a large portion of reasoning can be reused if we can identify and store these thoughts.

**(O2) Retrieval is faster than generation.** Retrieving information from a vector database is much faster than asking a language model to generate new text. While generating tokens requires sequential decoding on GPUs, retrieval simply fetches already-stored thoughts from memory. As shown in Figure 2 (middle), fetching from a large repository takes only a few milliseconds, whereas generating even a short piece of text takes seconds. Since output tokens are also several times more expensive than inputs, retrieval is both cheaper and faster.

**(O3) Reusing cached steps reduces tokens.** When correct steps are cached and reused, the number of tokens that the model must generate decreases substantially, as shown in Figure 2 (right). This is important because output tokens are both slower and more costly than input tokens. While adding retrieved steps slightly increases the prompt length, the savings in output tokens are much larger. As a result, reuse improves both cost and latency on datasets where problems share similar reasoning.

Together, these observations show that many reasoning steps are repeatable, retrieval is significantly more efficient than generation, and caching directly cuts down on token usage. This motivates our design of a system that can store and reuse thoughts to make reasoning more efficient. Full experimental details supporting these observations are provided in Appendix B.

## 4 RETRIEVAL-OF-THOUGHT

### 4.1 PRELIMINARIES

Before introducing our RoT framework, we formalize the key objects used throughout the paper: templates and reasoning steps, the graph structure that encodes their relationships, and the procedure for constructing semantic connections between steps. These preliminaries establish the foundation for representing and organizing reusable reasoning behaviors across problem-solving trajectories.

**Definition 1** (**Templates and Steps**). *Let $\mathcal{D}$ denote a finite set of problem–solving templates. Each template $t \in \mathcal{D}$ is an ordered sequence of reasoning steps $S^t := \left(s_0^t, s_1^t, \ldots, s_{l_{t-1}}^t\right)$, where $l_t \in \mathbb{N}$ is the template length. For index $i \in \{0, \ldots, l_{t-1}\}$, the pair $(t, i)$ denotes the $i$-th step of template $t$.*

**Definition 2** (**Thought Graph**). *The structure of reusable reasoning steps is encoded as a directed, weighted, disconnected multi-graph*

$$\mathcal{G} := \left(\mathcal{V}, \mathcal{E}_{Sequential} \cup \mathcal{E}_{Semantic}, w\right),$$

*where $\mathcal{V}$ is the set of nodes, $\mathcal{E} \subseteq \{(v_1, v_2) | (v_1, v_2) \in \mathcal{V}\}$, $w : \mathcal{E} \to \mathbb{R}$ is a weight function. Each node $(t, i) \in \mathcal{V}$ corresponds to a reasoning step from template $t$ at index $i$. Specifically, the set $\mathcal{V} = \{(t, i) : t \in \mathcal{D}, 0 \leq i < l_t\}$.*

The edge set $\mathcal{E}$ is partitioned into two disjoint types:

$$\mathcal{E} = \mathcal{E}_{\text{Sequential}} \cup \mathcal{E}_{\text{Semantic}}.$$

**Sequential Edges.** The set $\mathcal{E}_{\text{Sequential}}$ contains the sequential edges, which capture the within-template ordering of steps. For every $t \in \mathcal{D}$ and every pair of adjacent steps $(i, i+1)$ within it, we include a directed edge:

$$\mathcal{E}_{\text{Sequential}} = \left\{ \left((t, i), (t, i+1)\right) : t \in \mathcal{D}, \ 0 \leq i < l_{t-2} \right\}.$$

**Semantic Edges.** The set $\mathcal{E}_{\text{Semantic}}$ contains the semantic edges, which connect semantically analogous steps across different steps and templates. These edges are defined as a subset:

$$\mathcal{E}_{\text{Semantic}} = \left\{ \left((t, i), (t', j)\right) \in \mathcal{V} \times \mathcal{V} : (t, i) \neq (t', j), \ \widetilde{\text{sim}}(u_{t,i}, u_{t',j}) \geq \tau \right\}.$$

where $u_{t,i}$ is the embedding of node $(t, i)$, $\tau \in (0, 1)$ is a similarity threshold, and $\widetilde{\text{sim}}$ is a normalized similarity measure (defined below). Semantic edges are treated as bidirectional: if $(t, i)$ connects to $(t', j)$, then $(t', j)$ also connects to $(t, i)$.

**Construction of Semantic Edges.** To construct semantic edges, we begin by measuring the similarity between node embeddings using cosine similarity. All embeddings $u_{t,i}$ are $\ell_2$-normalized so that cosine similarity reduces to the dot product $\langle u_{t,i}, u_{t',j} \rangle$, and we rescale the result to lie within $[0, 1]$ using $\widetilde{sim}(a, b) = \frac{sim(a,b)+1}{2}$. The similarity threshold $\tau \in (0, 1)$ decide which node pairs are semantically similar enough to be connected and we use $\tau = 0.85$ for our study. Specifically, we include a semantic edge between two distinct nodes $(t, i)$ and $(t', j)$ if their normalized similarity score exceeds the threshold:

To ensure bidirectional connectivity, we insert both directions of each edge: whenever a pair $(t, i), (t', j)$ satisfies the similarity condition, both $((t, i), (t', j))$ and $((t', j), (t, i))$ are added to the graph, each with the same weight.

**Edge Weights.** The edge weight function $w : \mathcal{E} \rightarrow \mathbb{R}$ is defined as

$$w\big((t, i), (t', j)\big) = \begin{cases} 1, & \text{if } ((t, i), (t', j)) \in \mathcal{E}_{\text{Sequential}}, \\ \widetilde{sim}(u_{t,i}, u_{t',j}), & \text{if } ((t, i), (t', j)) \in \mathcal{E}_{\text{Semantic}}. \end{cases}$$

In practice, to make this process scalable and efficient, we precompute and cache all embeddings.

## 4.2 THOUGHT GRAPH CONSTRUCTION

We construct a directed thought graph that captures problem-solving thoughts from a curated set of mathematical reasoning templates. Each node corresponds to a single reasoning thought. Nodes contain metadata such as the template type and knowledge tags along with the reasoning text. Our graph is built from 3.34k templates drawn from the ReasonFlux-v2 dataset[2]. We manually verified that none of the templates overlaps with benchmarks used in evaluation, to ensure fair assessment.

The graph contains two types of edges. Sequential edges connect consecutive steps within the same template, preserving the natural logical flow. Semantic edges connect analogous steps across templates, supporting cross-template knowledge transfer. To establish semantic edges, we generate step embeddings using the `jina-embeddings-v2-small-en` model (Günther et al., 2023) and compute cosine similarity. All embeddings are pre-computed and cached for efficiency, with data sanitization ensuring XML-safe graph serialization.

Given a reasoning query from the user and a pre-constructed thought graph, the RoT framework operates as a three-step process: (1) retrieval of the initial node, (2) reward guided-traversal, and (3) template integration into the model. These steps together enable RoT to dynamically assemble contextually relevant reasoning templates from prior knowledge while preserving logical flow. We discuss each step in detail in the following subsections. The overall procedure is summarized in Algorithm 1, and we discuss each step in detail in the following subsections.

## 4.3 RETRIEVAL OF THE INITIAL NODE

The first step in constructing a reasoning template is to select an appropriate entry node from the thought graph. This is performed through a multi-stage filtering and scoring procedure.

**Filtering.** Candidate nodes are first restricted by metadata: only nodes whose template type matches the problem class (e.g., algebraic or geometric) are considered. Knowledge tags are then used to further refine the candidate set, retaining only nodes annotated with domain-specific concepts such as geometry, calculus, or number theory. Finally, validity checks ensure that each candidate node possesses a pre-computed embedding, guaranteeing consistency in similarity evaluation.

In our experiments, we manually annotated metadata tags (e.g., algebraic, geometric) for AIME and AMC problems. While this process could be automated using smaller encoder-only models such as BERT, well known for their effectiveness in text classification (Peng et al., 2021; Piękos et al., 2021), we did not train a specialized classifier here, since the goal of this work is not to propose a new tagging model but to demonstrate the impact of retrieval and traversal given such tags.

---

[2]The full version of the ReasonFlux-v2 dataset can be accessed using the link: https://huggingface.co/datasets/Gen-Verse/ReasonFlux-V2-Template

---

**Algorithm 1** Retrieval-of-Thought (RoT) Inference

---

1: **Input:** Query $\mathcal{Q}$; Thought Graph $\mathcal{G} = (\mathcal{V}, \mathcal{E}_{\text{Sequential}} \cup \mathcal{E}_{\text{Semantic}}, w)$; Reasoning Model $\mathcal{M}$
2: **System executes:**
3: Run metadata and knowledge-tag filtering on $\mathcal{V}$ to obtain candidate set of nodes.
4: Compute first-step reward $R_{\text{Initial}}$ (Equation 1); select $v_0 = \arg\max R_{\text{Initial}}$
5: Initialize template $\mathbf{T} \leftarrow [v_0]$
6: **while** termination condition is not met **do**
7:     Evaluate neighbors of current node using R (Equation 2)
8:     Select $v^* = \arg\max R$
9:     Append $v^*$ to T
10: **end while**
11: Construct prompt $\mathcal{P} = \langle \mathcal{Q}, \mathbf{T} \rangle$ with `<think>` tags
12: Call $\mathcal{M}$ API with $\mathcal{P}$: $\hat{y} \leftarrow \mathcal{M}(\mathcal{P})$
13: **return** $\hat{y}$

---

**Scoring.** The initial node is selected by maximizing a reward function that combines semantic relevance and structural validity:

$$R_{\text{Initial}} = \alpha \cdot R_Q + (1 - \alpha) \cdot R_S, \tag{1}$$

where $R_Q$ is the cosine similarity between the problem query and the node embedding, and $R_S$ is a structural indicator reward defined as

$$R_S = \begin{cases} 1, & \text{if } v_{(t,i)} \in \mathcal{V}, \ i = 0, \\ 0, & \text{otherwise.} \end{cases}$$

This criterion encourages the selection of nodes that are both semantically aligned with the query and structurally consistent as valid entry points. In our implementation, we set $\alpha = 0.8$ to prioritize semantic relevance while retaining a bias toward legitimate step-0 nodes.

### 4.4 REWARD-GUIDED TRAVERSAL

After initialization, the reasoning template is expanded through iterative graph traversal. At each step, candidate neighbors of the current node are evaluated by a reward function that balances semantic alignment with sequential flow.

**Graph Traversal.** The traversal reward is defined as

$$R = R_Q + R_F \tag{2}$$

where $R_Q$ denotes the query–semantic relevance of the candidate node, and $R_F$ is a structural flow reward given by

$$R_F = \begin{cases} 1, & \text{if } \big((t,i), (t,i{+}1)\big) \in \mathcal{E}_{\text{Sequential}}, \\ 0, & \text{otherwise.} \end{cases}$$

The traversal reward R assign equal weight to semantic alignment and structural consistency. Weighting factors can be introduced to adjust the traversal to emphasize either query-semantic relevance or structural flow.

**Template Termination.** Traversal halts when template expansion is no longer beneficial. Termination is also defined through a reward-based criterion, where continuation is penalized if quality or structural limits are violated: $(\max(R) < \tau) \lor (l_{\text{Template}} \geq l_{\text{max}}) \lor (N_{\text{Candidates}} = 0)$ where $l_{\text{Template}}$ is the current template length, $l_{\text{max}} = 8$ is the maximum allowed length, and $N_{\text{Candidates}}$ is the number of valid candidate nodes. This termination reward ensures that traversal halts when relevance falls below threshold, templates grow too long, or no candidates remain, thereby preventing unnecessary expansion and maintaining reasoning quality.

The specific parameter choices for thresholds and weighting factors are justified through sensitivity analysis in Appendix C.

### 4.5 TEMPLATE INTEGRATION

Once a template is retrieved, it is directly inserted into the prompt without additional processing. Prior work has shown that reasoning models often fail to reliably follow explicit instructions (Fu et al., 2025; Li et al., 2025a). To ensure that the model adheres to the retrieved reasoning during its internal deliberation, we place the template inside the `<think>` and `<\think>` tags, as introduced by Thinking Intervention (TI; Wu et al. (2025)). Importantly, we did not finetune the model specifically for template adherence, as such finetuning could risk catastrophic forgetting of core reasoning abilities. Empirical validation in Section 5 demonstrates that this simple integration is effective and usable in guiding the model's reasoning process.

## 5 EVALUATION

### 5.1 EVALUATION SETUP

We evaluate our approach on four mathematical reasoning benchmarks: AIME 2025, AIME 2024, AIME 2023, and AMC 2023, following the standard evaluation for reasoning techniques (Zhang et al., 2024; Snell et al., 2024; Yang et al., 2025b). We focus exclusively on mathematical datasets to enable reusability of a single thought graph across all evaluations, taking advantage of the consistent problem-solving patterns inherent to mathematics. Although our method is domain-agnostic, with potential for broader applications, we defer details on other domain reusability to Appendix B.

Our experiments use five models from the Qwen3 family: 0.6B, 1.7B, 4B, 8B, and 14B parameters. We choose Qwen3 for its *thinking mode*, which separates reasoning and output tokens using structured `<think>` and `</think>` tags. We evaluate two variants of our method: **RoT**, where the retrieved template is directly placed in the prompt, and **RoT+TI**, where the template is inserted inside the `<think>` tags following the Thinking Intervention mechanism. This distinction allows us to assess thought reuse both at the prompt level and within the model's internal reasoning process. All experiments are conducted with vLLM for efficient batch inference on NVIDIA A100 GPUs (40GB). For state-of-the-art comparison, we include CoT (Wei et al., 2022), CoT-SC (Wang et al., 2022), RAG (short for RAG+CoT), where we retrieve static templates same as those used to construct our Thought Graph combined with CoT, and BoT (Yang et al., 2024a).

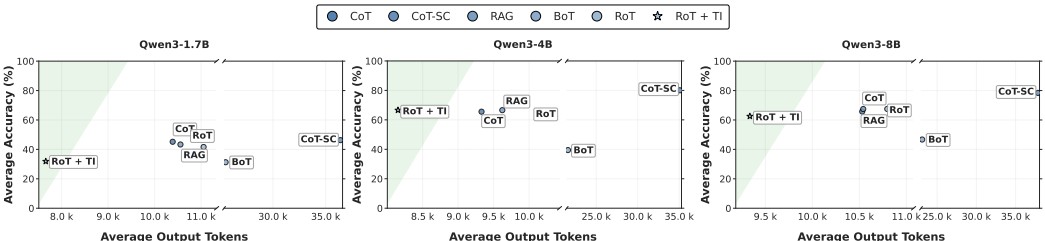

Figure 3: Average accuracy versus output tokens across Qwen3 models (1.7B, 4B, 8B). Each panel reports CoT, CoT-SC, RAG, BoT, RoT, and RoT+TI (star). The shaded top-left region denotes the *Efficient Reasoning Zone*, corresponding to higher accuracy with fewer tokens. RoT+TI consistently lies within this region, matching the accuracy of other methods while using way fewer tokens.

### 5.2 PERFORMANCE AND EFFICIENCY TRADE-OFFS

Figure 3 reports the trade-off between reasoning accuracy and token efficiency across Qwen3 models (1.7B, 4B, 8B). We find that RoT+TI consistently achieves comparable or better accuracy than baselines while substantially reducing output tokens, positioning it inside the *Efficient Reasoning Zone*. On Qwen3-1.7B, RoT+TI reaches roughly 44% accuracy versus 46% for CoT, while cutting output length by nearly 3k tokens (8.0k vs. 11.0k). At 4B scale, RoT+TI sustains 72% accuracy compared to 74% for CoT while lowering token usage by about 800 tokens (9.1k vs. 9.9k). Standard RoT also outperforms RAG by achieving similar accuracy with about 400 fewer tokens. On the larger 8B model, RoT+TI delivers 83% accuracy, slightly higher than RAG (80%), while using nearly 500 fewer tokens (9.6k vs. 10.1k). Importantly, the comparison between RAG and

RoT+TI underscores the benefits of dynamic templates: RoT+TI leverages reusable but context-sensitive structures, which generalize better than static templates in RAG or BoT, validating our hypothesis that dynamic template integration offers broader applicability when retrieving reasoning solutions. These results highlight that RoT+TI not only preserves reasoning performance but also systematically cuts down token usage across scales, demonstrating that integrating templates within <think> tags provides a robust pathway to efficient reasoning. Detailed results across all models and datasets, along with input–output example, are provided in Table 7 and Appendix D.

**Thought Graph Scalability Analysis.** We further verified scalability by comparing RoT with a smaller subgraph of randomly sampled 0.9k templates against the full 3.34k-template graph (Figure 10). Results show that increasing the number of templates generally improves accuracy across Qwen3 models, with particularly large gains at higher model scales (e.g., +17.0% on Qwen3-8B). These findings underscore that RoT is highly scalable: as LLM serving platforms accumulate more user data, the resulting larger thought graphs can yield systematically higher reasoning performance.

## 5.3 TOKEN USAGE AND COST IMPLICATIONS

We next analyze how thought reuse affects token distribution and monetary cost. While RoT slightly increases the number of input tokens due to template insertion, it consistently reduces output tokens by roughly 20%, yielding meaningful efficiency gains. Figure 4 reports the average per-sample inference cost across Qwen3 models using Alibaba Cloud token pricing and measured input/output token counts. RoT+TI delivers the largest savings at smaller scales, cutting costs by 59.0% on Qwen3-0.6B and 39.3% on Qwen3-1.7B relative to CoT. At mid-scale, the gains remain notable, with reduc-

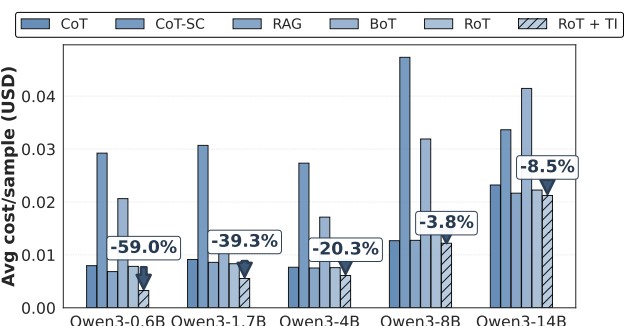

Figure 4: Average per-sample inference cost (USD) across Qwen3 models, comparing CoT, Cot-SC, RAG, BoT, RoT, and RoT+TI. Costs are computed using Alibaba Cloud prices with average input/output token over AIME 2023/2024/2025 and AMC 2023. Arrows above RoT+TI indicate the percent cost reduction relative to CoT.

tions of 20.3% on 4B and 3.8% on 8B. Even at 14B parameters, RoT+TI still achieves an 8.5% reduction. These results highlight that the efficiency benefits of thought reuse are strongest at small and medium scales but remain consistently positive across all model sizes. Full token and cost breakdowns, including pricing details, are provided in Tables 5 and 6 (Appendix D).

## 5.4 GENERALIZATION ACROSS MODEL FAMILIES

To test whether RoT is specific to the Qwen3 family, we evaluate RoT on two additional reasoning models from different model families: DLER-7B Liu et al. (2025) and DeepSeek-R1-Llama8B Guo et al. (2025). We keep the same RoT pipeline and evaluate on two math benchmarks (AMC 2023 and AIME 2024). As shown in Table 1, RoT and RoT+TI remain competitive across both model families. RoT often improves accuracy sometimes at the cost of longer outputs, while RoT+TI frequently recovers token efficiency and yields Pareto-efficient points on the accuracy–efficiency trade-off.

| Model | Data | Method | Acc. (%) | Out Tok. ($\Delta$ vs CoT) |
|-------|------|--------|----------|---------------------------|
| DLER-7B | AMC'23 | CoT | 85.00 | 1302.16 (–) |
| | | RoT | 90.00 | 1595.75 (+22.5%) |
| | | RoT+TI | 90.00 | 1077.53 (**–17.3%**) |
| | AIME'24 | CoT | 37.04 | 2167.78 (–) |
| | | RoT | 37.04 | 2434.22 (+12.3%) |
| | | RoT+TI | 51.85 | 2008.33 (**–7.3%**) |
| DS-R1-L8B | AMC'23 | CoT | 77.50 | 4372.45 (–) |
| | | RoT | 87.50 | 5419.40 (+23.9%) |
| | | RoT+TI | 72.50 | 2949.72 (**–32.6%**) |
| | AIME'24 | CoT | 40.74 | 9582.00 (–) |
| | | RoT | 51.85 | 9397.11 (**–1.9%**) |
| | | RoT+TI | 40.74 | 8647.00 (**–9.8%**) |

Table 1: **Generalization across model families** on AMC'23 and AIME'24. **DS-R1-L8B** abbreviates DeepSeek-R1-Llama8B.

## 5.5 GENERALIZATION BEYOND MATHEMATICS

While our main evaluation emphasizes math benchmarks for maximal reuse of a single thought graph, we also evaluate RoT on a non-math, scientific reasoning benchmark: GPQA. We construct a scientific thought graph from GPQA-style explanations and apply the same RoT procedure. Table 2 shows that RoT remains effective outside mathematics for mid/large models. RoT matches or improves accuracy while reducing output tokens, while RoT+TI yields particularly large token savings achieving up to ∼80%, highlighting that the efficiency benefits extend to domains where reasoning is less formulaic than math. This setting stresses whether RoT can reuse abstract reasoning moves (e.g., selecting relevant evidence, eliminating distractors, and composing a structured explanation) rather than relying on repeated algebraic patterns. Overall, these results suggest RoT is not inherently tied to mathematical formalism, but instead leverages transferable reasoning structure across domains.

| Model | Method | Acc. (%) | Out Tok. (△ vs CoT) |
|---|---|---|---|
| Qwen3-0.6B | CoT | 27.94 | 4039.28 (–) |
| | RoT | 29.41 | 4158.04 (+2.9%) |
| | RoT+TI | 20.59 | 797.34 (**−80.3%**) |
| Qwen3-1.7B | CoT | 30.23 | 5615.28 (–) |
| | RoT | 25.58 | 5496.93 (**−2.1%**) |
| | RoT+TI | 30.23 | 1448.26 (**−74.2%**) |
| Qwen3-4B | CoT | 58.21 | 5443.16 (–) |
| | RoT | 61.19 | 4773.43 (**−12.3%**) |
| | RoT+TI | 53.73 | 3021.03 (**−44.5%**) |
| Qwen3-8B | CoT | 53.97 | 5169.84 (–) |
| | RoT | 57.14 | 4165.19 (**−19.4%**) |
| | RoT+TI | 47.62 | 1646.27 (**−68.2%**) |
| Qwen3-14B | CoT | 57.69 | 3999.04 (–) |
| | RoT | 59.62 | 3336.25 (**−16.6%**) |
| | RoT+TI | 42.31 | 1954.94 (**−51.1%**) |

Table 2: Experimental results of RoT and RoT+TI on GPQA scientific-reasoning results.

## 5.6 ROBUSTNESS OF RoT ON NEAR-DUPLICATE VARIANTS

A potential concern for retrieval-based reasoning is confounding under near-duplicate inputs: two problems may share the same structure but differ in numerical constants, raising the possibility that retrieval could inject incorrect values. RoT avoids this failure mode because nodes in the thought graph encode *abstract reasoning operations* (e.g., "apply substitution" or "reduce to a standard form") rather than copying concrete numeric instantiations. To stress-test this setting, we construct a new near-duplicate dataset by generating structurally identical AIME and AMC variants via systematic changes to constants/bounds, build a thought graph using only these variants, and evaluate RoT on the original problems from AIME and AMC. Table 3 shows that RoT remains robust and RoT+TI provides substantial token reductions, demonstrating that RoT benefits further when reasoning structure repeats across queries.

| Model | Method | Acc. (%) | Out Tok. (△ vs CoT) |
|---|---|---|---|
| Qwen3-0.6B | CoT | 24.24 | 7916.44 (–) |
| | RoT | 31.82 | 7827.91 (**−1.1%**) |
| | RoT+TI | 21.21 | 3384.47 (**−57.2%**) |
| Qwen3-1.7B | CoT | 59.09 | 7747.24 (–) |
| | RoT | 66.67 | 7131.58 (**−7.9%**) |
| | RoT+TI | 50.00 | 4991.08 (**−35.6%**) |
| Qwen3-4B | CoT | 86.36 | 6754.41 (–) |
| | RoT | 83.33 | 6818.23 (+0.9%) |
| | RoT+TI | 80.30 | 5087.35 (**−24.7%**) |
| Qwen3-8B | CoT | 77.27 | 7368.03 (–) |
| | RoT | 89.39 | 7371.17 (+0.04%) |
| | RoT+TI | 77.27 | 5828.44 (**−20.9%**) |

Table 3: Experimental results of testing RoT with Near-duplicate AIME/AMC questions.

## 5.7 PATH SWITCHING ANALYSIS

How does RoT reduce unnecessary exploration during reasoning? We measure this effect by analyzing *path switching*, defined as the number of times a model abandons one reasoning trajectory and begins another within a single response. In practice, we approximate path switches by counting discourse markers that typically indicate a change of direction in reasoning, such as `"however"`, `"alternatively"`, or `"instead"`. Figure 5 reports the average number of path switches across Qwen3 models. We find that CoT frequently revises its trajectory, leading to longer outputs and wasted tokens. By contrast, RoT+TI reduces path switching by up to 81.8% (Qwen3-0.6B) and consistently lowers it

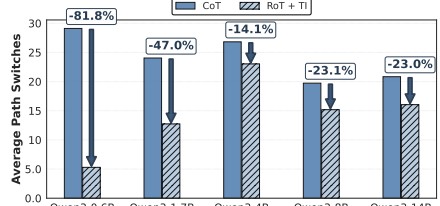

Figure 5: Average path switches across Qwen3 models comparing CoT with RoT+TI. RoT+TI consistently reduces unnecessary path exploration, achieving up to 81.8% fewer switches.

across larger models as well. These results confirm that template-guided reasoning anchors the model to promising trajectories, minimizing detours and thereby improving token efficiency.

## 5.8 LATENCY AND RETRIEVAL OVERHEAD ANALYSIS

We further examine the end-to-end latency of RoT, decomposing the cost into three components: retrieval and traversal, prefill, and decoding. While introducing retrieval incurs a small constant overhead, the overall effect is strongly positive, as reduced output tokens lead to significantly faster decoding. Figure 6 shows average per-sample latency across Qwen3 models. RoT+TI achieves the largest reductions at smaller scales, with latency drops of 72.9% on Qwen3-0.6B and 29.7% on Qwen3-1.7B compared to CoT. At mid-scale, latency decreases by 15.6% on 4B

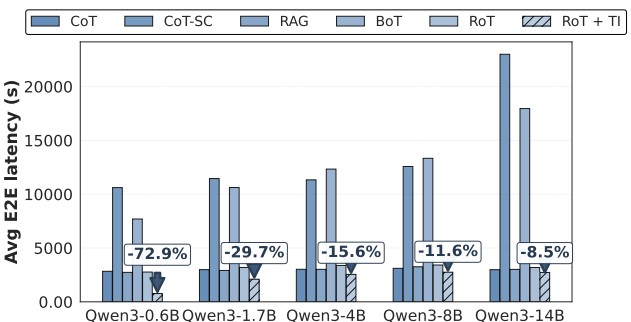

Figure 6: Average end-to-end latency (seconds) per sample across Qwen3 models, comparing CoT, Cot-SC, RAG, BoT, RoT, and RoT+TI. Arrows above RoT+TI indicate the percent latency reduction relative to CoT. RoT+TI achieves large gains on small and medium models (upto 82% reduction).

and 11.6% on 8B, while even at 14B, RoT+TI still yields an 8.5% reduction. These results highlight that retrieval-guided reasoning scales efficiently: although retrieval adds a minor front-end cost, the savings from reduced decoding dominate, yielding consistent latency improvements across all model sizes. A detailed breakdown of retrieval and traversal overheads is provided in Appendix D.

**Retrieval Overhead.** To quantify retrieval cost, we measured the latency of retrieval and traversal. As reported in Table 8 (Appendix D), retrieval takes on average only 0.038 s per query, with all models falling in the narrow range of 0.034 – 0.044 s. This overhead is negligible relative to the time spent on decoding, and thus does not contribute to total latency. Instead, the substantial reductions in output tokens dominate, making RoT+TI consistently faster than baseline methods.

## 6 DISCUSSION

**How much GPU memory overhead does the RoT have?** On an NVIDIA A100 (40 GB) with Qwen3-4B and batch size 12, the dominant consumers are KV cache and batching (25.0 GB) and LM weights (8.0 GB). The Thought Graph and the embedding model occupies only 1.7 GB, about 4.3% of total memory. Since the graph is stored once and shared across queries, its footprint remains small across models and batch sizes. Thus, the added cost is negligible compared to the KV+Batch bottleneck, confirming that RoT can be integrated without straining GPU memory budgets.

**Why do smaller models benefit more from RoT?** We find that smaller models exhibit larger gains with RoT, likely due to their stronger instruction-following ability. Since GRPO fine-tuning is typically performed on instruction-tuned bases, smaller models trained for fewer GRPO epochs, retain more of their supervised instruction-following behavior. Larger models, in contrast, undergo more extensive reinforcement training that amplifies exploration and raw reasoning ability but diminishes adherence to external templates, even after prompt tuning (Nimmaturi et al., 2025). This trade-off between reasoning and instruction following has been observed in prior studies (Fu et al., 2025), and explains why RoT's guidance is more effective at smaller scales. Looking forward, we believe efficiency gains will extend to larger models as future reasoning-focused checkpoints better preserve instruction-following alongside reasoning capacity.

## 7 CONCLUSION

We presented RoT, a framework that reuses prior reasoning through a thought graph and reward-guided traversal to construct problem-specific templates. RoT achieves substantial efficiency gains, reducing output tokens by up to 40%, latency by 82%, and inference cost by 67.5% while preserving accuracy across benchmarks. These results demonstrate that retrieval-guided template reuse offers a practical solution to the growing cost of reasoning-heavy inference. Looking forward, RoT provides a scalable foundation for efficient reasoning that extends beyond mathematics to broader domains.

## ETHICS STATEMENT

Our work focuses exclusively on mathematical reasoning datasets (AIME 2023/2024/2025, AMC 2023), which do not involve human subjects, personal data, or sensitive attributes. The proposed Retrieval-of-Thought framework is designed to improve inference-time efficiency and reduce compute and cost, with no foreseeable negative societal impacts such as privacy risks, misuse of personal information, or harmful deployment. We have carefully avoided data contamination by ensuring no overlap between the templates used in the thought graph and the benchmarks used for evaluation.

## REPRODUCIBILITY STATEMENT

We have taken deliberate steps to ensure the reproducibility of our results. All datasets used in this study are publicly available, and the construction process of the Thought Graph, including metadata annotation, semantic edge building, and caching procedures, is fully documented in the paper and appendices. Detailed algorithm descriptions, hyperparameter settings, and evaluation protocols are provided, along with sensitivity analyses for key parameters. We will release code, configuration files, and processed templates to enable researchers to reproduce and extend our findings.

## ACKNOWLEDGMENTS

The work of Azal Ahmad Khan was supported in part by the Amazon Machine Learning Systems Fellowship and the UMN GAGE Fellowship. The work of Shend Di was supported by the U.S. Department of Energy, Office of Science, Advanced Scientific Computing Research (ASCR), under contracts DE-AC02-06CH11357. The work of Ali Anwar was supported by the Samsung Global Research Outreach Award and the National Science Foundation Privacy-Preserving Data Sharing in Practice (PDaSP) program under grant number 2452817.

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

## A    RELATED WORKS

**Test-Time Scaling.**    Scaling test-time compute has emerged as a prominent strategy for enhancing the reasoning capabilities of LLMs (Snell et al., 2024). This approach mimics the human tendency to dedicate more cognitive effort to more challenging problems (Li et al., 2025b). Early implementations of this idea included methods like BoN sampling and various search-based strategies that explore multiple reasoning paths (Yao et al., 2023; Xie et al., 2023; Jiang et al., 2024a; Wang et al., 2025; Younsi et al., 2025). More recently, since the introduction of models like DeepSeek-R1, the focus has shifted towards RL, particularly using techniques like GRPO for fine-tuning (Guo et al., 2025). These methods encourage models to engage in multiple steps of self-reflection, leading to the generation of a huge number of output tokens during inference Bensal et al. (2025). By producing more extensive outputs, these models scale their compute at test-time, often leading to improved performance on complex tasks. However, this increased compute introduces two critical challenges: latency and cost (Cheshmi et al., 2025). The sequential generation of numerous tokens results in significantly higher latency, making real-time applications difficult. Furthermore, the associated costs are substantial, as providers typically price output tokens at a higher rate than input tokens, rendering these methods prohibitively expensive for widespread use (Han et al., 2024).

**Efficient Reasoning.**    High compute and latency overheads of reasoning workloads is creating an urgent need for inference-time efficiency. Recent works address this challenge through adaptive compute. Recent works that tries to parallelize inference achieving higher accuracy at the same latency compared to purely serialized reasoning approaches (Pan et al., 2025; Macfarlane et al., 2025; Ko et al., 2025; Chen et al., 2025; Xu et al., 2025b). At the architectural level, methods introduce token-level decision modules that selectively skip computation for less informative tokens, reducing FLOPs and KV-cache (Jiang et al., 2024b). Other approaches have addressed it by finetuning model to minimize unnecessary computational load, including techniques that adaptively select reasoning modes and early termination strategies that halt the thinking process once sufficient information has been generated Fang et al. (2025); Jiang et al. (2025); Zhang et al. (2025a).

**Retrieval Augmented Language Models.**    Retrieval-Augmented LMs (RALMs) have recently been explored as a way to inject external information into the reasoning process. BoT (Yang et al., 2024a) proposed storing distilled reasoning templates and retrieving them at inference time guide problem solving. SuperCorrect extended this by distilling multi-granular thought templates from larger teacher models and using them to supervise smaller student models (Yang et al., 2024b). Retrieval-Augmented Thoughts (RAT; Wang et al. (2024)) advanced this by iteratively revising chains with retrieved exemplars, improving long-horizon reasoning tasks such as math and program synthesis. Other works such as Chain of Evidences (CoE; Parvez (2024)) and ReARTeR (Sun et al., 2025) refine retrieval by grounding generation in evidential sequences or process rewards.

**Our Approach.**    However, prior approaches primarily target instruction-tuned models and rely on static templates that are only useful when the new problem closely resembles a previously solved variant. Such a static design restricts adaptability and limits efficiency gains. In contrast, our work focuses on models already equipped with strong reasoning capabilities, aiming to make their inference more efficient. Rather than storing entire templates, we decompose them into reusable thought steps that act as atomic reasoning primitives. Given a new query, our system dynamically assembles these steps into a fresh template tailored to the problem. This dynamic, fine-grained reuse makes our method substantially more flexible and widely applicable than previous retrieval-based approaches.

## B    ADDITIONAL MOTIVATION RESULTS

**(O1) Similarity of steps across datasets.**    We measured how similar reasoning steps are across a range of datasets. First, we solved the datasets using several LMs, prompted for stepwise answers, and parsed the outputs into discrete steps. For each step, we computed an embedding using `jina-embeddings-v2-small-en` model and compared it against stored thoughts in the thought graph using cosine similarity. We retrieved the top-10 most similar thoughts per step and used the best-match score to summarize similarity. As shown in Figure 2 (left), datasets such as AIME and AMC exhibit strong overlap in the 40–80% similarity range, confirming that many solution steps are reused across problems.

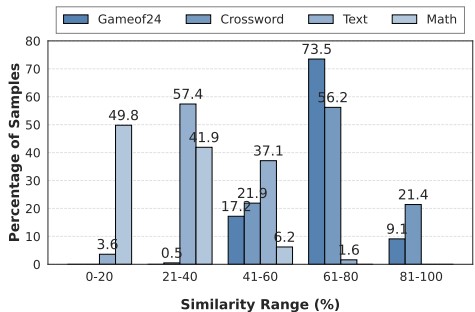

Figure 7: Semantic similarity of steps to solve reasoning questions across datasets.

We further confirmed this trend in additional datasets, shown in Figure 3. Here, tasks like Gameof24 and Crossword display high concentrations in the 61–80% range (73.5% and 56.2% of samples respectively), while Text and Math datasets skew toward lower ranges but still exhibit non-trivial similarity. These results demonstrate that reasoning problems across diverse domains frequently reuse similar thoughts, reinforcing that a substantial portion of reasoning can be efficiently supported through retrieval.

**(O2) Retrieving relevant information from a large vector database is significantly faster than generating text from a language model.** Reasoning models generate large token volumes. In the QWQ-LONGCOT-500K dataset, QwQ-32B averages 2,185 tokens per response, with 75% exceeding 8,000, driving up costs and latency as output tokens cost 4x more than input. Thought reuse reduces costs, as storage and DRAM are far cheaper than GPU-based token generation. Cached thoughts can be swiftly retrieved, fetching five matches from a 10,000-document vector DB takes 0.02s. In contrast, a Qwen2.5-7B generates 128 tokens in 1.62s, making retrieval quite faster than new tokens generation.

**(O3) Correctly cached reasoning steps reduce the reasoning tokens required for generation.** Thought reuse cuts output tokens by up to 42%, aligning with past studies. Our preliminary results from Figure 2 (right) show its effectiveness varies by dataset, high-similarity tasks like Gameof24 and Crossword benefit more than independent text problems. Thought reusability drives our project, as caching slightly increases prompt size but significantly reduces costlier output tokens ( 4x input cost). Since input tokens process in parallel while output tokens are sequential, reducing them also lowers latency. These results confirm that reusing thoughts improves both cost and efficiency.

## C  PARAMETER SELECTION DETAILS

We justify each parameter of RoT introduced in Section 4 using analysis experiments on the thought graph that we used in the paper. We compute query–step similarities using the `jinaai/jina-embeddings-v2-small-en` sentence embeddings (with L2 normalization).

**Semantic Edge Threshold.** We sweep the semantic threshold $\tau$ that determines which semantic edges are retained in the thought graph. The mean semantic degree (y-axis) vs. $\tau$ (x-axis) exhibits a clear knee (Figure 8) meaning the connectivity remains stable for $\tau \in [0.80, 0.85]$ at roughly 3 - 4 semantic neighbors per node, then drops sharply for $\tau \geq 0.87$. Setting $\tau = 0.85$ therefore places us at the knee preserving strong cross-template connectivity (useful for transfer) without flooding the graph with low-quality edges.

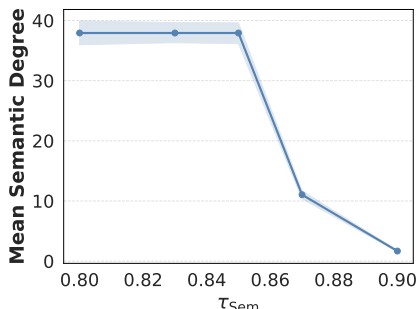

Figure 8: Average number of semantically similar nodes for different thresholds.

**First-Step Weights.** We quantify the first-step trade-off by plotting, as functions of $\alpha$, (i) the probability that the selected start lies at a true step-0 node and (ii) the average query–node similarity of the chosen start (Figure 9). On the full graph, these curves are smooth and monotonic, as $\alpha$ increases, semantic alignment consistently improves while

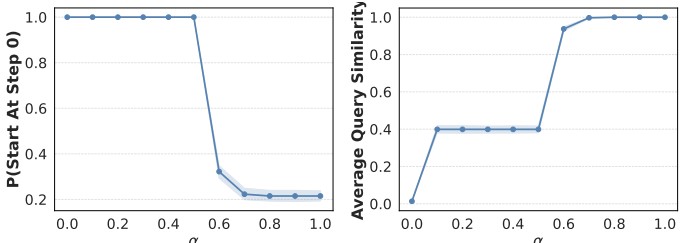

Figure 9: First-step selection trade-off on the Thought Graph. **Left** Probability that the chosen start is a at a step-0 node. **Right** Average query–node similarity of the chosen start.

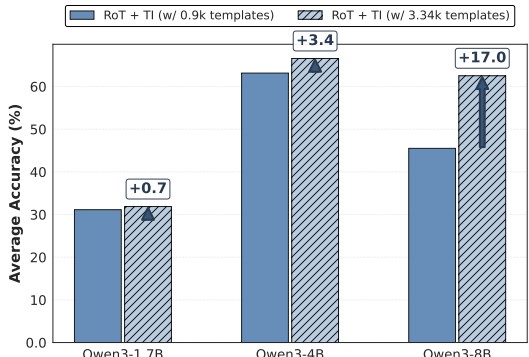

Figure 10: Template scalability analysis of RoT+TI. Accuracy is reported using a smaller thought graph with 0.9k templates (solid bars) versus the full 3.34k templates (hatched bars). Increasing the number of templates consistently improves performance, with larger models (e.g., Qwen3-8B) showing the greatest gains.

the bias toward legitimate starts reliably decreases. Notably, beyond $\alpha$=0.8 the marginal gains in similarity taper, whereas the loss in step-0 probability becomes more pronounced. We therefore set $\alpha$=0.8, which delivers a high-alignment starting point without discarding the structural prior to begin at valid template starts, and we observe that this choice is stable with tight uncertainty bands under full-graph sampling.

# D  ADDITIONAL MAIN RESULTS

"OOT" stands for "Out of Tokens," indicating that the model was unable to generate the answer within the maximum number of tokens permitted for its response.

## D.1  ACCURACY AND EFFICIENCY RESULTS

### D.1.1  TEMPLATE GRAPH SCALABILITY ANALYSIS

To examine the scalability of our framework, we conducted an ablation where the thought graph was constructed using only 0.9k templates randomly sampled from our dataset, compared to the full 3.34k-template graph used in the main experiments. Figure 10 reports the accuracy of RoT+TI under both settings across Qwen3-1.7B, 4B, and 8B models. The results indicate that accuracy consistently increases as the number of templates grows, with particularly pronounced improvements at larger model sizes. For example, Qwen3-8B improves by 17.0 percentage points when moving from 0.9k to 3.34k templates. These results demonstrate that RoT scales favorably with graph size, suggesting that platforms with access to abundant user reasoning traces can expect significant gains from larger template repositories.

Table 4: Performance comparison across different LLMs and baseline methods

| Method | AIME 2025 | | AIME 2024 | | AIME 2023 | | AMC 2023 | |
|---|---|---|---|---|---|---|---|---|
| | Acc (%) | Tokens | Acc (%) | Tokens | Acc (%) | Tokens | Acc (%) | Tokens |
| **Qwen3-0.6B** | | | | | | | | |
| CoT | 6.67 | 10004.93 | 3.7 | 13533.03 | 13.33 | 10421.53 | 42.5 | 6447.14 |
| RAG | 13.33 | 10523.86 | 7.41 | 12533.04 | 10 | 10849.5 | 40 | 7037.12 |
| BoT | 6.67 | 38792.17 | 3.7 | 35882 | 13 | 30258.34 | 25 | 21943.69 |
| CoT-SC | 13.33 | 44520.4 | 11.11 | 47558.93 | 16.67 | 42095.83 | 40 | 26191.12 |
| **RoT** | 6.67 | 11994 | 11.11 | 11952.29 | 10 | 10933.53 | 35 | 7388.15 |
| **RoT + TI** | 6.67 | 5277.87 | 0 | 4318.7 | 0 | 3058.8 | 15 | 2289.92 |
| **Qwen3-1.7B** | | | | | | | | |
| CoT | 26.67 | 13111.87 | 40.74 | 11525.89 | 33.33 | 11676.13 | 80 | 6403.09 |
| RAG | 26.67 | 13280.8 | 33.33 | 12230.67 | 33.33 | 11456.13 | 80 | 6842.54 |
| BoT | 13.33 | 33657.47 | 25.93 | 41947.96 | 23.33 | 45898.9 | 62.5 | 50371.8 |
| CoT-SC | 20 | 47804.8 | 51.85 | 30307.18 | 33.33 | 45657.83 | 80 | 25292.25 |
| **RoT** | 20 | 12791.73 | 40.74 | 11436.88 | 33.33 | 12577.16 | 72.5 | 8478.95 |
| **RoT + TI** | 20 | 8568.47 | 33.33 | 9173.96 | 16.67 | 8292.7 | 57.5 | 5955.47 |
| **Qwen3-4B** | | | | | | | | |
| CoT | 53.33 | 11488.33 | 55.56 | 10076.15 | 63.33 | 11366.2 | 90 | 5550.17 |
| RAG | 60 | 11670.26 | 62.96 | 10335.7 | 53.33 | 11829.9 | 90 | 6240.79 |
| BoT | 26.67 | 50194.53 | 14.81 | 43174.37 | 36.4 | 49331.67 | 80 | 35440.58 |
| CoT-SC | 80 | 42002.8 | 77.78 | 38140.92 | 70 | 44736.6 | 92.5 | 19142.35 |
| **RoT** | 60 | 11634.6 | 66.67 | 10621.33 | 46.67 | 13277.83 | 87.5 | 6133.01 |
| **RoT + TI** | 66.67 | 9421 | 62.96 | 9481.59 | 46.67 | 10003.3 | 90 | 5052.9 |
| **Qwen3-8B** | | | | | | | | |
| CoT | 60 | 12770.6 | 62.96 | 11232.55 | 50 | 12812.63 | 90 | 6468.07 |
| RAG | 66.67 | 11777.46 | 55.56 | 12283.48 | 60 | 12728.53 | 87.5 | 6960.72 |
| BoT | 33.33 | 59970.86 | 18.52 | 49993.15 | 50 | 56428.9 | 85 | 40784.82 |
| CoT-SC | 80 | 43571.8 | 77.78 | 41127.48 | 63.33 | 49529.87 | 92.5 | 20963.55 |
| **RoT** | 60 | 12831.67 | 66.67 | 11561.92 | 53.33 | 13341.63 | 90 | 6522.53 |
| **RoT + TI** | 53.33 | 11279.07 | 59.26 | 10930.03 | 50 | 10698.93 | 87.5 | 5780.75 |
| **Qwen3-14B** | | | | | | | | |
| CoT | 73.33 | 9451.27 | 77.78 | 9685.22 | 60 | 12096.93 | 97.5 | 5669.67 |
| RAG | 86.67 | 9910.86 | 66.67 | 10981.38 | 73.33 | 11750.4 | 95 | 5277.07 |
| BoT | OOT | OOT | OOT | OOT | OOT | OOT | OOT | OOT |
| CoT-SC | 66.67 | 39272.87 | 74.07 | 36955.67 | 63.33 | 46998.97 | 97.5 | 17032.82 |
| **RoT** | 80 | 10187 | 66.67 | 11078.44 | 56.67 | 12430.8 | 92.5 | 6114.88 |
| **RoT + TI** | 66.67 | 9767.33 | 62.96 | 11201.03 | 63.33 | 10623.03 | 82.5 | 5191.57 |

Table 5: Performance comparison across different LLMs and baseline methods.

| Method | AIME 2023 | | | AIME 2024 | | | AIME 2025 | | | AMC 2023 | | |
|---|---|---|---|---|---|---|---|---|---|---|---|---|
| | In Tok | Out Tok | Cost ($) | In Tok | Out Tok | Cost ($) | In Tok | Out Tok | Cost ($) | In Tok | Out Tok | Cost ($) |
| **Qwen3-0.6B** | | | | | | | | | | | | |
| CoT | 285.6 | 9719.33 | $0.18 | 279.59 | 13253.44 | $0.45 | 335.4 | 10086.13 | $0.38 | 260.57 | 6186.57 | $0.31 |
| RAG | 399.33 | 10124.53 | $0.19 | 386 | 12147.04 | $0.41 | 428.6 | 10420.9 | $0.40 | 371.27 | 6665.85 | $0.34 |
| BoT | 8234.67 | 30557.5 | $0.59 | 6530.7 | 29351.3 | $1.02 | 6342.87 | 23915.47 | $0.92 | 5604.96 | 16338.73 | $0.85 |
| CoT-SC | 906.13 | 43614.27 | $0.83 | 900 | 46658.93 | $1.59 | 1105.73 | 40990.1 | $1.55 | 806.5 | 25384.62 | $1.28 |
| **RoT** | 275.8 | 11718.2 | $0.22 | 273.44 | 11678.85 | $0.40 | 304.83 | 10628.7 | $0.40 | 216.03 | 7172.12 | $0.36 |
| **RoT + TI** | 345.8 | 4932.07 | $0.09 | 343.59 | 3975.11 | $0.14 | 372.9 | 2685.9 | $0.10 | 286.02 | 2003.9 | $0.10 |
| **Qwen3-1.7B** | | | | | | | | | | | | |
| CoT | 285.6 | 12826.27 | $0.24 | 279.59 | 11246.3 | $0.38 | 335.4 | 11340.73 | $0.43 | 260.57 | 6142.52 | $0.31 |
| RAG | 399.33 | 12881.47 | $0.24 | 386 | 11844.67 | $0.40 | 428.6 | 11027.53 | $0.42 | 371.27 | 6471.27 | $0.33 |
| BoT | 17598.55 | 16058.92 | $0.33 | 15296.48 | 26651.48 | $0.95 | 18988.8 | 26910.1 | $1.08 | 17918.07 | 32453.73 | $1.71 |
| CoT-SC | 919.13 | 46885.67 | $0.89 | 598.74 | 29708.44 | $1.01 | 1142.5 | 44515.33 | $1.69 | 806.5 | 24485.75 | $1.24 |
| **RoT** | 275.8 | 12515.93 | $0.24 | 273.44 | 11163.44 | $0.38 | 304.83 | 12272.33 | $0.46 | 216.03 | 8262.92 | $0.42 |
| **RoT + TI** | 345.8 | 8222.67 | $0.16 | 343.59 | 8830.37 | $0.30 | 372.9 | 7919.8 | $0.30 | 286.02 | 5669.45 | $0.29 |
| **Qwen3-4B** | | | | | | | | | | | | |
| CoT | 285.6 | 11202.73 | $0.21 | 279.59 | 9796.56 | $0.33 | 335.4 | 11030.8 | $0.42 | 260.57 | 5289.6 | $0.27 |
| RAG | 399.33 | 11270.93 | $0.21 | 386 | 9949.7 | $0.34 | 428.6 | 11401.3 | $0.43 | 371.27 | 5869.52 | $0.30 |
| BoT | 25746.73 | 24447.8 | $0.50 | 22015.56 | 21158.81 | $0.79 | 22563.67 | 26768 | $1.09 | 21432.03 | 14008.55 | $0.80 |
| CoT-SC | 920.67 | 41082.13 | $0.78 | 889.48 | 37251.44 | $1.27 | 1114.3 | 43622.3 | $1.65 | 806.5 | 18335.85 | $0.93 |
| **RoT** | 275.8 | 11358.8 | $0.22 | 273.44 | 10347.89 | $0.35 | 304.83 | 12973 | $0.49 | 216.03 | 5916.98 | $0.30 |
| **RoT + TI** | 345.8 | 9075.2 | $0.17 | 343.59 | 9138 | $0.31 | 372.9 | 9630.4 | $0.37 | 286.02 | 4766.88 | $0.24 |
| **Qwen3-8B** | | | | | | | | | | | | |
| CoT | 285.6 | 12485 | $0.39 | 279.59 | 10952.96 | $0.62 | 335.4 | 12477.23 | $0.79 | 260.57 | 6207.5 | $0.52 |
| RAG | 399.33 | 11378.13 | $0.36 | 386 | 11897.48 | $0.68 | 428.6 | 12299.93 | $0.78 | 371.27 | 6589.45 | $0.56 |
| BoT | 32881.13 | 27089.73 | $0.94 | 25573.15 | 24420 | $1.51 | 29494.1 | 26934.8 | $1.86 | 26854.97 | 13929.85 | $1.36 |
| CoT-SC | 920.67 | 42651.13 | $1.35 | 899.78 | 40227.7 | $2.29 | 1142.5 | 48387.37 | $3.05 | 806.5 | 20157.05 | $1.70 |
| **RoT** | 275.8 | 12555.87 | $0.40 | 273.44 | 11288.48 | $0.64 | 304.83 | 13036.8 | $0.82 | 216.03 | 6306.5 | $0.53 |
| **RoT + TI** | 345.8 | 10933.27 | $0.35 | 343.59 | 10586.44 | $0.60 | 372.9 | 10326.03 | $0.65 | 286.02 | 5494.73 | $0.46 |
| **Qwen3-14B** | | | | | | | | | | | | |
| CoT | 285.6 | 9165.67 | $0.58 | 279.59 | 9405.63 | $1.07 | 335.4 | 11761.53 | $1.49 | 260.57 | 5409.1 | $0.91 |
| RAG | 277.13 | 9633.73 | $0.61 | 271.19 | 10710.19 | $1.22 | 326.8 | 11423.6 | $1.44 | 252.55 | 5024.52 | $0.85 |
| BoT | OOT | OOT | | OOT | OOT | OOT | OOT | OOT | OOT | OOT | OOT | |
| CoT-SC | 920.67 | 38352.2 | $2.42 | 891.41 | 36064.26 | $4.10 | 1098.57 | 45900.4 | $5.79 | 769.52 | 16263.3 | $2.74 |
| **RoT** | 275.8 | 9911.2 | $0.63 | 273.44 | 10805 | $1.23 | 304.83 | 12125.97 | $1.53 | 216.03 | 5898.85 | $0.99 |
| **RoT + TI** | 345.8 | 9421.53 | $0.60 | 343.59 | 10857.44 | $1.23 | 372.9 | 10250.13 | $1.30 | 286.02 | 4905.55 | $0.83 |

## D.2 TOKEN USAGE AND COST IMPLICATIONS

The cost shows the total amount to run the entire experiment.

## D.3 INFERENCE API COSTS

Table 6: Inference API Costs for Qwen3 models Alibaba Cloud.

| Model | Input Price (Million Tokens) | Output Price (Million Tokens) |
|---|---|---|
| Qwen3-0.6b | $0.11 | $1.26 |
| Qwen3-1.7b | $0.11 | $1.26 |
| Qwen3-4b | $0.11 | $1.26 |
| Qwen3-8b | $0.18 | $2.10 |
| Qwen3-14b | $0.35 | $4.20 |
| Qwen3-32b | $0.70 | $8.40 |

## D.4 LATENCY RESULTS

Table 7: Performance comparison across different LLMs and baseline methods

| Method | AIME 2023 Latency (s) | AIME 2024 Latency (s) | AIME 2025 Latency (s) | AMC 2023 Latency (s) |
|---|---|---|---|---|
| **Qwen3-0.6B** | | | | |
| CoT | 86.7 | 171.01 | 116.85 | 68.05 |
| RAG | 94.75 | 157.17 | 127.03 | 79.68 |
| BoT | 298.48 | 337.85 | 270 | 174.42 |
| CoT-SC | 509.71 | 611.43 | 542.92 | 343.21 |
| **RoT** | 120.11 | 141.02 | 129.02 | 79.73 |
| **RoT + TI** | 32.93 | 29.58 | 17.07 | 12.7 |
| **Qwen3-1.7B** | | | | |
| CoT | 163.69 | 150.75 | 163 | 76.71 |
| RAG | 165.17 | 165 | 156.58 | 81.16 |
| BoT | 197.6 | 345.94 | 369.73 | 438.35 |
| CoT-SC | 627.12 | 401.14 | 672.93 | 337.36 |
| **RoT** | 154.66 | 148.85 | 176.77 | 110.95 |
| **RoT + TI** | 79.28 | 112.52 | 98.87 | 64.05 |
| **Qwen3-4B** | | | | |
| CoT | 182.6 | 166.24 | 195.11 | 78.58 |
| RAG | 186.69 | 172.58 | 206.86 | 91.1 |
| BoT | 455.45 | 410.22 | 560.21 | 268.87 |
| CoT-SC | 715.87 | 692.94 | 870.46 | 317.24 |
| **RoT** | 184.52 | 177.89 | 245.36 | 90.53 |
| **RoT + TI RoT + TI** | 135.33 | 158.86 | 163.43 | 73.8 |
| **Qwen3-8B** | | | | |
| CoT | 281.04 | 245.56 | 285.25 | 126.8 |
| RAG | 244.59 | 267.58 | 285.09 | 137.25 |
| BoT | 741.66 | 556.07 | 694.16 | 324.38 |
| CoT-SC | 910.9 | 900.96 | 1109.18 | 413.16 |
| **RoT** | 273.8 | 255.75 | 303.62 | 129.82 |
| **RoT + TI** | 246.3 | 240.14 | 228.72 | 113.79 |
| **Qwen3-14B** | | | | |
| CoT | 241.62 | 252.21 | 322.19 | 142.71 |
| RAG | 255.87 | 291.2 | 311.75 | 131.91 |
| BoT | OOT | OOT | OOT | OOT |
| CoT-SC | 1015.18 | 969.05 | 1255.63 | 420.81 |
| **RoT** | 263.15 | 293.53 | 331.5 | 155.12 |
| **RoT + TI** | 251.24 | 295.89 | 278.88 | 129.89 |

Table 8: Retrieval time (seconds) across datasets and Qwen3 models.

| Model | Qwen3-0.6B | Qwen3-1.7B | Qwen3-14B | Qwen3-4B | Qwen3-8B |
|---|---|---|---|---|---|
| AIME 2023 | 0.0435 | 0.0434 | 0.0385 | 0.0375 | 0.0403 |
| AIME 2024 | 0.0402 | 0.0406 | 0.0356 | 0.0338 | 0.0331 |
| AIME 2025 | 0.0505 | 0.0342 | 0.0294 | 0.0300 | 0.0423 |
| AMC 2023 | 0.0411 | 0.0376 | 0.0437 | 0.0356 | 0.0344 |
| Average | 0.0438 | 0.0389 | 0.0368 | 0.0342 | 0.0375 |

## D.5 PATH SWITCHING RESULTS

As shown in Table 9, RoT+TI promotes stable, single-path reasoning, cutting back on oscillations that inflate outputs under CoT. As summarized in the table, all Qwen3 models benefit, with path-switch reductions spanning 14.2–81.8%. The effect persists at larger scales, suggesting that template grounding complements scale rather than substituting for it. RoT+TI assists the model in guiding its

Table 9: Average path switches per output (CoT vs RoT + TI)

| Method | Avg Switches | △ vs CoT (%) |
|---|---|---|
| **Qwen3-0.6B** | | |
| CoT | 29.116 | – |
| **RoT + TI** | **5.295** | **-81.8** |
| **Qwen3-1.7B** | | |
| CoT | 24.045 | – |
| **RoT + TI** | **12.732** | **-47.1** |
| **Qwen3-4B** | | |
| CoT | 26.821 | – |
| **RoT + TI** | **23.027** | **-14.2** |
| **Qwen3-8B** | | |
| CoT | 19.732 | – |
| **RoT + TI** | **15.179** | **-23.1** |
| **Qwen3-14B** | | |
| CoT | 20.839 | – |
| **RoT + TI** | **16.045** | **-23.0** |

reasoning, enabling it to arrive at the answer by adhering to the template. An illustrative example can be observed in the blocks below, which depict the output from RoT+TI, RoT, and CoT, respectively. Through RoT+TI, the model successfully solved the questions using only 920 tokens, whereas RoT and CoT generated significantly more than 6000 tokens to solve the same question. This disparity arises due to the multiple path switching that occurs during the model's problem-solving process, as indicated by the (........) in the output. RoT retrieves the appropriate template for the problem, but the model fails to adhere to the template even when it is provided in the prompt. In contrast, RoT+TI enables the model to solve the problem without path switching.

---

**RoT+TI: Input and Output for AIME 2024 Q2 - No. Out Tokens: 920**

**Input:** You are given a template to solve the problem. Use the given steps if applicable otherwise use them to guide your reasoning and present the solution steps logically to solve this problem:

Problem: There exist real numbers $x$ and $y$, both greater than 1, such that $\log_x\left(y^x\right) = \log_y\left(x^{4y}\right) = 10$. Find $xy$. <|im_end|><|im_start|>assistant<*think*> I need to follow the given steps to guide me in reasoning to solve the problem. I will follow each step and modify the steps to make them match the problem and then solve the problem accordingly. The template is. Step 1: Take logarithms (base equal to one variable) of both sides to linearize exponents. Example: For $x^y = y^x$, take $\log_x$ to get $y = x\log_x y$. Strictly following the given steps for guidance, I will now solve the problem starting from the step 1. Using step 1

**Output:** Take logarithms (base equal to one variable) of both sides to linearize exponents. Let me apply this to the given equation. Given: $\log_x(y^x) = \log_y(x^{4y}) = 10$.

Step 1: Take logarithms (base equal to one variable) of both sides to linearize exponents. For the first equation, $\log_x(y^x) = 10$, I can rewrite this using logarithm properties. Recall that $\log_b(a^c) = c\log_b a$. So: $\log_x(y^x) = x\log_x y = 10$. Similarly, for the second equation $\log_y(x^{4y}) = 10$, apply the same property: $\log_y(x^{4y}) = 4y\log_y x = 10$. Now, I have two equations:

1. $x\log_x y = 10$
2. $4y\log_y x = 10$

Let me denote $\log_x y = a$. Then $\log_y x = \frac{1}{a}$, since $\log_b a \cdot \log_a b = 1$. Substituting into the equations:

1. $x \cdot a = 10 \rightarrow a = \frac{10}{x}$
2. $4y \cdot \frac{1}{a} = 10 \rightarrow \frac{4y}{a} = 10$ Substituting $a = \frac{10}{x}$ into the second equation: $\frac{4y}{\frac{10}{x}} = 10$. Simplify: $\frac{4y \cdot x}{10} = 10$. Multiply both sides by 10: $4xy = 100$ Divide both sides by 4 $xy = 25$. So, the value of $xy$ is 25.

But wait, the problem states that $x$ and $y$ are both greater than 1. Let me check if this is consistent. From $xy = 25$, since both x and y are greater than 1, this is possible. For example, x = 5, y = 5. Both are greater than 1. So the solution is valid. Therefore, the answer is 25<*/think*>.

To solve the problem, we are given two logarithmic equations:

$$\log_x(y^x) = \log_y(x^{4y}) = 10$$

—

Step 1: Use logarithmic properties to simplify the expressions. We apply the logarithmic identity $\log_b(a^c) = c\log_b a$ to each equation.

First equation: $\log_x(y^x) = x\log_x y = 10$ Second equation: $\log_y(x^{4y}) = 4y\log_y x = 10$

—

Step 2: Introduce a substitution to simplify. Let $a = \log_x y$. Then by the change of base formula, $\log_y x = \frac{1}{a}$, since $\log_b a \cdot \log_a b = 1$. Now substitute into the equations:

First equation becomes:

$$x \cdot a = 10 \quad \Rightarrow \quad a = \frac{10}{x}$$

Second equation becomes:

$$4y \cdot \frac{1}{a} = 10 \quad \Rightarrow \quad \frac{4y}{a} = 10$$

Substitute $a = \frac{10}{x}$ into the second equation:

$$\frac{4y}{\frac{10}{x}} = 10 \quad \Rightarrow \quad \frac{4xy}{10} = 10$$

Multiply both sides by 10:

$$4xy = 100 \quad \Rightarrow \quad xy = 25$$

—

Final Answer:

$$\boxed{25}$$

---

**RoT: Input and Output for AIME 2024 Q2 - No. Out Tokens: 6746**

**Input:** "You are given a template to solve the problem. Use the given steps if applicable otherwise use them to guide your reasoning and present the solution steps logically to solve this problem: Step 1: Take logarithms (base equal to one variable) of both sides to linearize exponents. Example: For $x^y = y^x$, take $\log_x$ to get $y = x \log_x y$. Problem: There exist real numbers $x$ and $y$, both greater than 1, such that $\log_x (y^x) = \log_y (x^{4y}) = 10$. Find $xy$."

**Output:** *<think>*Okay, let's try to solve this problem. So, we have two equations here because both logarithms are equal to 10. The problem says that log base x of $y^x$ equals 10, and log base y of $x^{4y}$ also equals 10. And we need to find the product xy, where both x and y are greater than 1.First, I need to recall logarithm properties.........

Hmm. Now, I need to relate these two equations. Let me think about how to express $\log_x(y)$ and $\log_y(x)$.........

Let me recall that $\log_y(x)$ can be written using change of base formula: $\log_y(x) = ln(x)/ln(y)$. Alternatively, since we have expressions for x and y in terms of a, maybe we can use the definition of logarithm..........

Let me see if I can take logarithms of both sides to make it easier. Taking natural logarithm on both sides..........

Alternatively, since $log_x(y^x) = 10$, which is the same as $y^x = x^{10}$ because $log_x(A) = B \implies A = x^B$. So $y^x = x^{10}$. Similarly, $log_y(x^{4y}) = 10$ implies that $x^{4y} = y^{10}$. So now we have two equations:1) $y^x = x^{10}$ 2) $x^{4y} = y^{10}$ So maybe we can manipulate these equations. Let me take the first equation..........

I should have noticed that earlier instead of going through all the logarithmic manipulations and numerical solving. **Final Answer** $\boxed{25}$ *</think>*

To solve the problem: We are given the following equations involving logarithms:

$$\log_x(y^x) = \log_y(x^{4y}) = 10$$

with the constraint that $x > 1$ and $y > 1$, and we are to find the value of $xy$.

—

Step 1: Apply Logarithmic Identities using the logarithmic identity: $\log_b(a^c) = c \log_b(a)$
We rewrite both equations: 1. $\log_x(y^x) = x \log_x(y) = 10$ 2. $\log_y(x^{4y}) = 4y \log_y(x) = 10$

—

Step 2: Use Change of Base Formula
We know that: $\log_b(a) = \frac{1}{\log_a(b)}$ So, $\log_x(y) = \frac{1}{\log_y(x)}$. Let's define: $a = \log_y(x) \Rightarrow \log_x(y) = \frac{1}{a}$
Now substitute into the first equation:

$$x \cdot \frac{1}{a} = 10 \Rightarrow x = 10a$$

Substitute into the second equation:

$$4y \cdot a = 10 \Rightarrow y = \frac{10}{4a} = \frac{5}{2a}$$

—

Step 3: Use the Definition of Logarithms
From the definition of logarithms, $a = \log_y(x)$ implies: $y^a = x$ Substitute the expressions for $x$ and $y$: $\left(\frac{5}{2a}\right)^a = 10a$ This equation is transcendental and difficult to solve algebraically, but we can proceed to find $xy$ without solving for $x$ and $y$ explicitly.

—

Step 4: Compute $xy$ From earlier, we found: $x = 10a, \quad y = \frac{5}{2a}$ So, $xy = (10a) \cdot \left(\frac{5}{2a}\right) = \frac{50a}{2a} = 25$

—

Conclusion
Thus, regardless of the specific values of $x$ and $y$, as long as they satisfy the original logarithmic equations, the product $xy$ is always:

$$\boxed{25}$$

---

**CoT: Input and Output for AIME 2024 Q2 - No. Out Tokens: 6022**

**Input:** "Solve the following math problem efficiently and clearly. Present the solution steps logically.
- For complex problems (3 steps or more): Use this step-by-step format:
Step 1: [Concise description] [Brief explanation and calculations]
Step 2: [Concise description] [Brief explanation and calculations]
...
Regardless of the approach, always conclude with:
Therefore, the final answer is: $\boxed{answer}$.
Where [answer] is just the final numerical answer that solves the problem. Ensure the number is clearly identifiable within the box.
Problem: There exist real numbers $x$ and $y$, both greater than 1, such that $\log_x(y^x) = log_y(x^{4y}) = 10$. Find $xy$.
Let's solve this step-by-step "

**Output:**<*think*> Okay, let's try to solve this problem. So, we have two logarithmic equations: log base x of $y^x$ equals 10, and log base y of $x^{4y}$ also equals 10. And we need to find the value of xy, where x and y are both greater than First, I remember that logarithms have some properties that might help here..........

Wait, so that's straightforward? So, the answer is 25? Let me check if that makes sense. Let me verify. If xy = 25, then let's see if the equations hold. Suppose xy = 25. Let me pick some numbers. Let me assume..........

But then why when I tried x=5, y=5, it didn't work? Because maybe x and y don't have to be integers. Let me think..........

Wait, $25^2$ is 625, divided by 4 is 156.25. So, 156.25 = $(25/2)^2$. So, $\log_2((25/2)^2) = 2 log_2(25/2) = 2[\log 2_(25) - \log_2(2)] = 2[\log_2(25) - 1]$. $\log_2(25)$ is $\log_2(5^2) = 2\log_2(5)$. So, $2[2\log_2(5) - 1] = 4\log_2(5) - 2$. But $\log_2(5)$ is approximately 2.3219, so 4*2.3219 = 9.2876 - 2 = 7.2876. Which is not 10. So, that's not 10. Therefore, something is wrong here. But according to the algebra..........

Let me let x = 5, then y=5. But then, check the first equation: $y^x = 5^5 = 3125, x^{10} = 5^10 = 9765625$. Not equal. So, not equal. Alternatively, let me take x = sqrt(25) = 5, but same as above. Wait, maybe x and y are not integers. Let me think..........

Therefore, the equations are consistent, and the previous contradiction was due to an error in substitution. Therefore, the solution is indeed xy=25. Therefore, the answer is 25. So, even though finding the exact values of x and y is complicated, the algebra shows that xy=25 is the correct answer. Therefore, the final answer is $\boxed{25}$ </*think*> We are to find the value of $xy$, given that both $x$ and $y$ are real numbers greater than 1.

---

Step 1: Use the logarithmic identity We apply the logarithmic identity:

$$\log_b(a^c) = c\log_b(a)$$

This gives us: 1. $\log_x(y^x) = x \cdot \log_x(y) = 10$ 2. $\log_y(x^{4y}) = 4y \cdot \log_y(x) = 10$ Let's denote: - $\log_x(y) = \frac{\ln y}{\ln x} = \frac{10}{x}$, from the first equation.- $\log_y(x) = \frac{\ln x}{\ln y} = \frac{5}{2y}$, from the second equation.

---

Step 2: Use the reciprocal relationship From the first equation:

$$\log_x(y) = \frac{10}{x} \Rightarrow \log_y(x) = \frac{x}{10}$$

From the second equation:

$$\log_y(x) = \frac{5}{2y}$$

Setting the two expressions for $\log_y(x)$ equal: $\frac{x}{10} = \frac{5}{2y}$

---

Step 3: Solve for $xy$ Cross-multiplying: $2xy = 50 \Rightarrow xy = 25$

---

Final Answer

$$\boxed{25}$$

