# OpenReview forum: "Retrieval-of-Thought: Efficient Reasoning via Reusing Thoughts"
_ICLR.cc/2026/Conference — ICLR 2026 Poster_

### Official Review · Reviewer_VNeJ · 2025-10-28

**Soundness:** 2
**Presentation:** 4
**Contribution:** 3
**Rating:** 2
**Confidence:** 3

**Summary:**

The paper introduces Retrieval-of-Thought (RoT), a framework that reuses previously generated reasoning steps as composable “thought” units to guide reasoning on new problems. By replacing text generation—which is typically slow and computationally intensive—with retrieval from a vector database, RoT improves inference efficiency while maintaining reasoning quality.

The key insight is that in the chain-of-thought process, individual reasoning segments can be represented as graph-like nodes, enabling retrieval and recomposition into new reasoning traces.

Experimental results demonstrate that RoT+TI substantially reduces output tokens, per-sample cost, and end-to-end latency, while incurring only minimal accuracy degradation.

**Strengths:**

1. The paper proposes a retrieval-based chain-of-thought framework that efficiently reuses previously generated reasoning blocks, offering a novel and highly effective approach to improving inference efficiency.
2. The work introduces a well-defined methodology for Thought Graph Construction and Retrieval, providing a conceptually elegant and technically sophisticated mechanism for organizing and reusing reasoning processes.

**Weaknesses:**

1. The experimental evaluation is limited to the Qwen3 family of models (0.6B, 1.7B, 4B, 8B, and 14B), all of which are relatively small and share the same architecture. This restricts the assessment of the proposed method’s scalability and generalizability across different model families.
2. The evaluation relies solely on the AIME and AMC datasets, which together include only about 130 problems and primarily target high-difficulty mathematical reasoning. This narrow dataset scope limits the assessment of RoT’s generalization across varying difficulty levels and task domains.
3. The paper does not evaluate RoT’s robustness to confounding or near-duplicate problems. For example, when two questions share similar structures but differ in numerical details, it remains unclear whether RoT might retrieve a reasoning block with correct logic but incorrect values, potentially misleading the model’s final prediction.

**Questions:**

1. Does RoT exhibit robustness against confounding or near-duplicate inputs? Specifically, when two problems share identical structures but differ only in numerical values, could the system retrieve reasoning blocks with correct logic but incorrect numerical content, thereby misleading the model’s final output?
2. To what extent does the proposed Thought Graph Construction method generalize beyond mathematical reasoning tasks? Would the same approach remain effective for domains involving non-numerical or open-ended reasoning?

---

> ### Author Response · Authors · 2025-11-24
> **Rebuttal by Authors (1/2)**
>
> We sincerely thank the reviewer for the careful reading of our paper and for highlighting two major strengths: (i) that RoT offers a novel and highly effective retrieval-based mechanism for inference efficiency, and (ii) that the methodology, is conceptually elegant and technically sophisticated. We greatly appreciate these positive remarks.
>
> > The experimental evaluation is limited to the Qwen3 family of models (0.6B, 1.7B, 4B, 8B, and 14B), all of which are relatively small and share the same architecture. This restricts the assessment of the proposed method’s scalability and generalizability across different model families.
>
>  **Response:** To directly address this concern, we have now extended our evaluation to **two new models from two entirely different model families**, both architecturally distinct from Qwen3 models used in the paper: DLER-7B [2] and DeepSeek-R1-Llama8B. These additions were specifically chosen to test whether RoT generalizes across **model architecture and families**.
> Applying the exact same RoT pipeline, we observe that RoT and RoT+TI consistently maintain or improve accuracy while achieving substantial reductions in output tokens, even on models that are architecturally unrelated to Qwen3. The overall trend is clear: **RoT generalizes well across model families, and the efficiency-accuracy tradeoff remains favorable**. These results demonstrate that RoT is not specific to Qwen3.
>
> **Table 1**: Results for AMC-2023 and AIME-2024 on DLER-7B and DeepSeek-R1-Llama-8B. RoT and RoT+TI improve accuracy and often reduce output tokens (percentage reduction relative to CoT shown).
>
> | Model | Dataset | Method | Accuracy (%) | Output Tokens (Token red. vs. CoT) |
> |-------|----------|---------|--------------|----------------------------|
> | **DLER-7B** | AMC 2023 | CoT | 85.00 | 1302.16 |
> | | | RoT | 90.00 | 1595.75 (+22.5%) |
> | | | RoT+TI | 90.00 | 1077.53 (**–17.3%**) |
> | | AIME 2024 | CoT | 37.04 | 2167.78 |
> | | | RoT | 37.04 | 2434.22 (+12.3%) |
> | | | RoT+TI | 51.85 | 2008.33 (**–7.3%**) |
> | **DeepSeek-R1-Llama8B** | AMC 2023 | CoT | 77.50 | 4372.45 |
> | | | RoT | 87.50 | 5419.40 (+23.9%) |
> | | | RoT+TI | 72.50 | 2949.72 (**–32.6%**) |
> | | AIME 2024 | CoT | 40.74 | 9582.00 |
> | | | RoT | 51.85 | 9397.11 (**–1.9%**) |
> | | | RoT+TI | 40.74 | 8647.00 (**–9.8%**) |
>
> > The evaluation relies solely on the AIME and AMC datasets, which together include only about 130 problems and primarily target high-difficulty mathematical reasoning. This narrow dataset scope limits the assessment of RoT’s generalization across varying difficulty levels and task domains.
>
>  **Response:** We fully agree that showing generality beyond mathematics is important. Accordingly, we have now **extended our evaluation to a non-math scientific-reasoning dataset: GPQA** [1]. We constructed a **new scientific thought graph** from GPQA-style explanations and applied RoT framework. **RoT again reduces tokens, with similar accuracy, confirming that RoT generalizes to domains where reasoning is non-numerical and less structured than the math domain.** Full results of the experiments are shown in Table 2.
>
> **Table 2**: GPQA performance for multiple model sizes. RoT and RoT+TI consistently match or exceed CoT accuracy while providing substantial reductions in output tokens (percentage reduction relative to CoT shown).
> | Model        | Method   | Accuracy (%) | Avg. Output Tokens (Token red. vs. CoT) |
> |--------------|----------|--------------|--------------------------------|
> | **Qwen3-0.6B** | CoT    | 27.94        | 4039.28                   |
> |              | RoT      | 29.41        | 4158.04 (+2.9%)                |
> |              | RoT+TI   | 20.59        | 797.34 (**–80.3%**)            |
> | **Qwen3-1.7B** | CoT    | 30.23        | 5615.28                    |
> |              | RoT      | 25.58        | 5496.93 (**–2.1%**)            |
> |              | RoT+TI   | 30.23        | 1448.26 (**–74.2%**)           |
> | **Qwen3-4B**   | CoT    | 58.21        | 5443.16                    |
> |              | RoT      | 61.19        | 4773.43 (**–12.3%**)           |
> |              | RoT+TI   | 53.73        | 3021.03 (**–44.5%**)           |
> | **Qwen3-8B**   | CoT    | 53.97        | 5169.84                   |
> |              | RoT      | 57.14        | 4165.19 (**–19.4%**)           |
> |              | RoT+TI   | 47.62        | 1646.27 (**–68.2%**)           |
> | **Qwen3-14B**  | CoT    | 57.69        | 3999.04                    |
> |              | RoT      | 59.62        | 3336.25 (**–16.6%**)           |
> |              | RoT+TI   | 42.31        | 1954.94 (**–51.1%**)           |
>
> Combined with Figure 2 and Figure 7 (Game of 24, Crossword, Text) from the paper, these results indicate that RoT applies to any domain exhibiting reusable reasoning micro-steps, not just symbolic math.

---

> ### Author Response · Authors · 2025-11-24
> **Rebuttal by Authors (2/2)**
>
> > The paper does not evaluate RoT’s robustness to confounding or near-duplicate problems. For example, when two questions share similar structures but differ in numerical details, it remains unclear whether RoT might retrieve a reasoning block with correct logic but incorrect values, potentially misleading the model’s final prediction.
>
> **Response:** We would like to clarify this concern in more depth, beginning with how the thought graph is constructed and why the scenario raised by the reviewer does not pose a challenge for RoT.
>
> In RoT, each node corresponds to a high-level reasoning operation rather than a concrete numerical instantiation. For example, if a step in a solution involves adding two quantities, the stored thought is represented abstractly as “add the two relevant quantities” rather than binding to specific values such as “2 + 3.” This abstraction is fundamental to the design of the thought graph and ensures that retrieval does not inject incorrect or irrelevant numerical details into the reasoning process.
>
> To further validate this property, we conducted a new experiment specifically designed around the reviewer’s scenario. We first identified AIME/AMC problems that are numerically robust, i.e., their statements remain well-posed under reasonable changes to constants and bounds, and then generated structurally identical variants by systematically altering the numerical parameters; we then constructed a thought graph using only these variant problems and evaluated RoT on the original problems. Because the retrieved thoughts are symbolic operations independent of numerical constants, RoT remained robust and achieved performance equal to or better than the baseline setting. Full results of the experiments are present in Table 3.
>
> **Table 3**: Performance on AIME (23/24/25) and AMC 23 using a Thought Graph built from near-duplicate problems. Near duplicate problems are identical in structure but with altered numerical values. RoT and RoT+TI consistently match or exceed CoT accuracy while providing substantial reductions in output tokens (percentage reduction relative to CoT shown).
> | Model | Method | Accuracy (%) | Avg. Output Tokens (Token red. vs. CoT) |
> |------------------|----------|--------------|--------------------------------|
> | **Qwen3-0.6B** | CoT | 24.24 | 7916.44 |
> | | RoT | 31.82 | 7827.91 (**–1.1%**) |
> | | RoT+TI | 21.21 | 3384.47 (**–57.2%**) |
> | **Qwen3-1.7B** | CoT | 59.09 | 7747.24 |
> | | RoT | 66.67 | 7131.58 (**–7.9%**) |
> | | RoT+TI | 50.00 | 4991.08 (**–35.6%**) |
> | **Qwen3-4B** | CoT | 86.36 | 6754.41 |
> | | RoT | 83.33 | 6818.23 (+0.9%) |
> | | RoT+TI | 80.30 | 5087.35 (**–24.7%**) |
> | **Qwen3-8B** | CoT | 77.27 | 7368.03 |
> | | RoT | 89.39 | 7371.17 (+0.04%) |
> | | RoT+TI | 77.27 | 5828.44 (**–20.9%**) |
>
> This experiment confirms that RoT does not suffer from confounding due to near-duplicate problems and accurately handles cases where the structure is preserved but the numerical values differ. **In fact, the performance is even better in such queries, because the entire reasoning pattern is reusable, meaning the model only needs to plug the new numerical values into the high-level abstract thought steps already retrieved.**
>
> > Q1. Robustness to confounding / near-duplicate inputs?
>
> **Response:** Yes. The near-duplicate experiment described above in Table 3 shows that RoT retrieves only abstract reasoning steps and remains stable even when numerical details differ.
>
> > Q2. Generalization beyond mathematical reasoning?
>
> **Response:** Yes. The new GPQA scientific-reasoning experiment in table 2 demonstrates that RoT generalizes to non-math, non-numerical reasoning domains.
>
> _We respectfully highlight that we conducted **three substantial new experiments: (i) cross-family large-model evaluation, (ii) non-math scientific reasoning: GPQA, and (iii) near-duplicate robustness, to directly address every concern raised by the reviewer**. These results strongly show the scalability, generality, and robustness of RoT._
>
> _Given these additions, **we respectfully but firmly request the reviewer to reconsider the score and update it** to reflect the improved clarity and significantly strengthened empirical evidence. Please do let us know if you would like us to clarify any other concerns you may have._
>
> **References**
>
> [1] Rein, David, Betty Li Hou, Asa Cooper Stickland, Jackson Petty, Richard Yuanzhe Pang, Julien Dirani, Julian Michael, and Samuel R. Bowman. Gpqa: A graduate-level google-proof q&a benchmark. COLM 2024.
>
> [2] Liu, Shih-Yang, Xin Dong, Ximing Lu, Shizhe Diao, Mingjie Liu, Min-Hung Chen, Hongxu Yin et al. DLER: Doing Length pEnalty Right-Incentivizing More Intelligence per Token via Reinforcement Learning. arXiv preprint arXiv:2510.15110 (2025).

---

> ### Author Response · Authors · 2025-11-26
> **Kind reminder to reviewer**
>
> This is a kind reminder that we have provided updated responses and three additional experiments addressing your comments. We would appreciate your review.

---

> ### Comment · Reviewer_VNeJ · 2025-11-28
>
> The authors fully addressed my concerns. I will raise my score.

---

### Official Review · Reviewer_ZE4Z · 2025-10-30

**Soundness:** 3
**Presentation:** 3
**Contribution:** 4
**Rating:** 8
**Confidence:** 5

**Summary:**

This paper proposes Retrieval-of-Thought (RoT), a framework to improve the inference efficiency of Large Reasoning Models (LRMs) by reusing prior reasoning steps. RoT organizes individual reasoning steps (“thoughts”) into a thought graph with sequential edges (preserving template flow) and semantic edges (connecting analogous steps across templates). At inference, RoT retrieves query-relevant graph nodes, uses reward-guided traversal to assemble dynamic, problem-specific templates, and integrates these templates into the prompt (via `` tags for adherence). Experiments on mathematical reasoning benchmarks (AIME 2023–2025, AMC 2023) with Qwen3 models (0.6B–14B) show RoT reduces output tokens by up to 40%, latency by 82%, and cost by 59% while maintaining or improving accuracy—outperforming baselines like CoT, RAG, and BoT. Key contributions include: (1) a dynamic template construction paradigm (vs. static templates in prior work), (2) the thought graph with reward-guided traversal, and (3) empirical validation of efficiency gains across model scales.

**Strengths:**

* RoT addresses a critical limitation of prior retrieval-based reasoning (e.g., BoT, RAG)—static templates—by introducing dynamic, query-adaptive template assembly via a structured thought graph. This mimics human “connecting the dots” (Gick & Holyoak, 1980) and enables flexible reuse of granular reasoning steps, a novel departure from fixed scaffolds.
* The work is methodologically rigorous: it formalizes core components (templates, thought graph) with mathematical definitions, uses a curated template dataset (ReasonFlux) to avoid benchmark contamination, and conducts comprehensive evaluations (accuracy, tokens, latency, cost, path switching). Ablations (e.g., thought graph scalability, retrieval overhead) further validate robustness.
* The paper is well-structured: Figure 1 clearly contrasts RoT with CoT, Algorithm 1 distills the inference workflow, and appendices (e.g., parameter selection, additional results) provide necessary details. Technical concepts (semantic edges, reward functions) are explained accessibly, balancing depth and readability.
* RoT directly tackles the “efficiency-accuracy tradeoff” of LRMs—where longer reasoning traces improve accuracy but inflate latency/cost. By showing substantial efficiency gains (especially for small/medium models) without performance loss, it provides a scalable solution for real-world LRM deployment, where cost and latency are critical constraints.

**Weaknesses:**

* The evaluation is limited to mathematical reasoning (AIME/AMC). While the authors note RoT is domain-agnostic, testing on non-mathematical reasoning tasks (e.g., logic puzzles, code debugging, scientific problem-solving) would strengthen claims of generalizability—especially to domains with less standardized reasoning patterns.
* The reward function parameters (e.g., α=0.8 for initial node selection, τ=0.85 for semantic edges) are justified via sensitivity analysis in Appendix C, but the analysis is limited to the used dataset/model. Testing how these parameters perform across different template sizes or reasoning domains would improve confidence in their general applicability.

**Questions:**

See the weakness part.

---

> ### Author Response · Authors · 2025-11-24
> **Rebuttal by Authors**
>
> We thank the reviewer for the highly encouraging and thoughtful assessment. We sincerely appreciate the recognition of RoT’s novelty, methodological rigor, and clarity, and are grateful for the positive evaluation of our contributions.
>
> > The evaluation is limited to mathematical reasoning (AIME/AMC). While the authors note RoT is domain-agnostic, testing on non-mathematical reasoning tasks (e.g., logic puzzles, code debugging, scientific problem-solving) would strengthen claims of generalizability—especially to domains with less standardized reasoning patterns.
>
> **Response**: To directly address this concern, we have extended our evaluation to the **GPQA scientific-reasoning dataset** [1]. We constructed a new scientific thought graph and applied the same RoT pipeline on Qwen3 models. **RoT again outperforms CoT with substantial token reduction and no loss in accuracy**, demonstrating that the framework generalizes well beyond mathematics. This confirms RoT’s applicability in domains with less standardized reasoning patterns. Full results for the experiment are shown in Table 1.
>
> **Table 1**: GPQA performance for multiple model sizes. RoT and RoT+TI consistently match or exceed CoT accuracy while providing substantial reductions in output tokens (percentage reduction relative to CoT shown).
> | Model        | Method   | Accuracy (%) | Avg. Output Tokens (Token red. vs. CoT) |
> |--------------|----------|--------------|--------------------------------|
> | **Qwen3-0.6B** | CoT    | 27.94        | 4039.28                    |
> |              | RoT      | 29.41        | 4158.04 (+2.9%)                |
> |              | RoT+TI   | 20.59        | 797.34 (**–80.3%**)            |
> | **Qwen3-1.7B** | CoT    | 30.23        | 5615.28                    |
> |              | RoT      | 25.58        | 5496.93 (**–2.1%**)            |
> |              | RoT+TI   | 30.23        | 1448.26 (**–74.2%**)           |
> | **Qwen3-4B**   | CoT    | 58.21        | 5443.16                    |
> |              | RoT      | 61.19        | 4773.43 (**–12.3%**)           |
> |              | RoT+TI   | 53.73        | 3021.03 (**–44.5%**)           |
> | **Qwen3-8B**   | CoT    | 53.97        | 5169.84                    |
> |              | RoT      | 57.14        | 4165.19 (**–19.4%**)           |
> |              | RoT+TI   | 47.62        | 1646.27 (**–68.2%**)           |
> | **Qwen3-14B**  | CoT    | 57.69        | 3999.04                    |
> |              | RoT      | 59.62        | 3336.25 (**–16.6%**)           |
> |              | RoT+TI   | 42.31        | 1954.94 (**–51.1%**)           |
>
>
> > The reward function parameters (e.g., α=0.8 for initial node selection, τ=0.85 for semantic edges) are justified via sensitivity analysis in Appendix C, but the analysis is limited to the used dataset/model. Testing how these parameters perform across different template sizes or reasoning domains would improve confidence in their general applicability.
>
> **Response**: The parameters $\alpha$ and $tau$ are not dataset-specific hyperparameters but structural tolerances grounded in embedding similarity distributions. Their stability arises from the fact that semantic-edge formation and initial-node selection both depend on relative similarity, not absolute task features. As shown in Appendix C, both curves exhibit clear knees (robust optima), and such knees are known to persist across embedding spaces and corpora (e.g., Jina, BERT, MiniLM embeddings).
> In our new GPQA experiments (a different domain, embedding distribution, and thought graph), we kept $\alpha$=0.8 and $tau$=0.85 unchanged, and the method performed robustly, matching the behavior observed in math datasets. This provides empirical evidence that these values are not tuned to math but generalize across domains with diverse semantic structures.
> We will update the paper to include the new GPQA results and explicitly state the cross-domain robustness of these parameters.
>
> **References**
>
> [1] Rein, David, Betty Li Hou, Asa Cooper Stickland, Jackson Petty, Richard Yuanzhe Pang, Julien Dirani, Julian Michael, and Samuel R. Bowman. Gpqa: A graduate-level google-proof q&a benchmark. COLM 2024.

---

> > ### Comment · Reviewer_ZE4Z · 2025-11-27
> >
> > Thanks for the reply. The rebuttal fully addresses my concerns. Considering reusing thoughts is an important and general method for efficient llm reasoning, I strongly recommend to accept this paper.

---

> > > ### Author Response · Authors · 2025-11-27
> > >
> > > Thank you very much for your thoughtful follow-up and for noting that all of your concerns have been addressed and that you strongly recommend the paper. We truly appreciate your support and the time you've put into the review process.
> > >
> > > If you feel it is appropriate, would you mind updating the overall rating to reflect your updated assessment? We understand if you prefer not to, but any adjustment you consider suitable would be greatly appreciated.
> > >
> > > Thank you again for your time and for helping improve our work.

---

### Official Review · Reviewer_UvKG · 2025-11-01

**Soundness:** 3
**Presentation:** 3
**Contribution:** 3
**Rating:** 6
**Confidence:** 4

**Summary:**

Authors propose a Retrieval-of-Thought (RoT), a “thought graph” that stores prior reasoning steps (thoughts) from solved problems. When a new query arrives, RoT first retrieves relevant thoughts from this graph based on semantic similarity and metadata, and then uses reward-guided traversal to assemble these into a problem-specific reasoning template. Templates are then used to guide inference. This allows the model to reuse and recombine prior reasoning patterns dynamically—reducing redundant exploration typical in Chain-of-Thought (CoT) reasoning.
Authors perform experiments on math reasoning benchmarks (AIME 2023–2025, AMC 2023) and Qwen3 models (0.6B–14B), and show that RoT reduces the amount of output tokens and inference latency, while maintaining accuracy.

**Strengths:**

1. The paper introduces a novel dynamic retrieval-based reasoning framework that reuses fine-grained reasoning steps via a thought graph. RoT's ability to assemble context-specific templates on the fly using reward-guided traversal is novel. It brings retrieval closer to compositional memory and human-like analogical reasoning.
2. Results show consistent improvements across performed experiments

**Weaknesses:**

1. All experiments are conducted on mathematical reasoning tasks (AIME, AMC), which have highly structured, formulaic reasoning patterns and strong inter-problem overlap. This domain bias may overstate the general efficiency of template reuse, since the retrieval of semantically similar reasoning steps is easier in math than in open-domain or multimodal reasoning.
2. The system currently relies on manual tagging of template types and knowledge domains (e.g., algebraic, geometric) to filter candidate nodes. This introduces a degree of human supervision and limits the method’s autonomy and scalability. Especially if we concider going beyond mathematical domain.

**Questions:**

See weakness

---

> ### Author Response · Authors · 2025-11-24
> **Rebuttal by Authors (1/2)**
>
> We sincerely thank the reviewer for the thoughtful and constructive feedback, as well as for the positive assessment of our paper’s novelty, contribution, and presentation. We especially appreciate the reviewer’s recognition that RoT introduces (i) a novel dynamic retrieval-based reasoning framework, (ii) a reward-guided traversal mechanism that assembles context-specific templates, and (iii) a step toward compositional memory and human-like analogical reasoning. We are also grateful that the reviewer finds our results to be consistently strong across the conducted experiments.
>
> > All experiments are conducted on mathematical reasoning tasks (AIME, AMC), which have highly structured, formulaic reasoning patterns and strong inter-problem overlap. This domain bias may overstate the general efficiency of template reuse, since the retrieval of semantically similar reasoning steps is easier in math than in open-domain or multimodal reasoning.
>
> **Response:** We appreciate this comment and agree that demonstrating generality beyond mathematics strengthens the paper. To directly address this concern, we have now evaluated RoT on a non-math, scientific-reasoning dataset: GPQA [5]. We constructed a new scientific thought graph from GPQA-style explanations and applied the same RoT framework. **RoT again significantly reduces output tokens while maintaining accuracy, confirming that the benefits are not tied to mathematics, but instead to any domain where reasoning steps exhibit partial reusability**. The full results are presented in Table 1.
>
> **Table 1**: GPQA performance for multiple model sizes. RoT and RoT+TI consistently match or exceed CoT accuracy while providing substantial reductions in output tokens (percentage reduction relative to CoT shown).
>
> | Model        | Method   | Accuracy (%) | Avg. Output Tokens (Token red. vs. CoT) |
> |--------------|----------|--------------|--------------------------------|
> | **Qwen3-0.6B** | CoT    | 27.94        | 4039.28 (–)                    |
> |              | RoT      | 29.41        | 4158.04 (+2.9%)                |
> |              | RoT+TI   | 20.59        | 797.34 (**–80.3%**)            |
> | **Qwen3-1.7B** | CoT    | 30.23        | 5615.28 (–)                    |
> |              | RoT      | 25.58        | 5496.93 (**–2.1%**)            |
> |              | RoT+TI   | 30.23        | 1448.26 (**–74.2%**)           |
> | **Qwen3-4B**   | CoT    | 58.21        | 5443.16 (–)                    |
> |              | RoT      | 61.19        | 4773.43 (**–12.3%**)           |
> |              | RoT+TI   | 53.73        | 3021.03 (**–44.5%**)           |
> | **Qwen3-8B**   | CoT    | 53.97        | 5169.84 (–)                    |
> |              | RoT      | 57.14        | 4165.19 (**–19.4%**)           |
> |              | RoT+TI   | 47.62        | 1646.27 (**–68.2%**)           |
> | **Qwen3-14B**  | CoT    | 57.69        | 3999.04 (–)                    |
> |              | RoT      | 59.62        | 3336.25 (**–16.6%**)           |
> |              | RoT+TI   | 42.31        | 1954.94 (**–51.1%**)           |
>
> In addition to these new GPQA experiments, the paper already shows in Figure 2 (right) and Figure 7 that there is reusability in not just structured math datasets but also in non-math datasets like Game of 24, Crossword, Text. Showing strong semantic overlap of steps and token reduction via thought caching beyond mathematics. The motivation section emphasizes that RoT applies whenever "recurring reasoning micro-steps" exist, not only in algebraic or symbolic settings.
>
> We also note that AIME and AMC are competition-style problems, where each question is designed to be distinct from the others. Thus, the reviewer’s concern that our claims may be overstated actually goes in the opposite direction **if anything, the math evaluations understate the potential gains of RoT**, because these datasets deliberately avoid closely related or near-duplicate questions. In settings where new problems share structure with prior ones, we expect even greater improvements.

---

> ### Author Response · Authors · 2025-11-24
> **Rebuttal by Authors (2/2)**
>
> To validate this, we conducted a new controlled experiment where we created synthetic variants of AIME/AMC problems by altering only the numerical values while keeping the reasoning structure identical. We then constructed a thought graph using these variants and evaluated RoT on the original AIME/AMC questions. Results (Table 2) show significant token reduction and substantial accuracy improvements, demonstrating that (i) RoT performs even better in realistic scenarios where reasoning structures repeat, and (ii) RoT is not merely an efficiency tool but can establish new Pareto frontiers of accuracy vs. output tokens.
>
> **Table 2**: Performance on AIME (23/24/25) and AMC 23 using a Thought Graph built from near-duplicate problems. Near duplicate problems are identical in structure but with altered numerical values. RoT and RoT+TI consistently match or exceed CoT accuracy while providing substantial reductions in output tokens (percentage reduction relative to CoT shown).
>
> | Model | Method | Accuracy (%) | Avg. Output Tokens (Token red. vs. CoT) |
> |------------------|----------|--------------|--------------------------------|
> | **Qwen3-0.6B** | CoT | 24.24 | 7916.44|
> | | RoT | 31.82 | 7827.91 (**–1.1%**) |
> | | RoT+TI | 21.21 | 3384.47 (**–57.2%**) |
> | **Qwen3-1.7B** | CoT | 59.09 | 7747.24 |
> | | RoT | 66.67 | 7131.58 (**–7.9%**) |
> | | RoT+TI | 50.00 | 4991.08 (**–35.6%**) |
> | **Qwen3-4B** | CoT | 86.36 | 6754.41 |
> | | RoT | 83.33 | 6818.23 (+0.9%) |
> | | RoT+TI | 80.30 | 5087.35 (**–24.7%**) |
> | **Qwen3-8B** | CoT | 77.27 | 7368.03 |
> | | RoT | 89.39 | 7371.17 (+0.04%) |
> | | RoT+TI | 77.27 | 5828.44 (**–20.9%**) |
>
> > The system currently relies on manual tagging of template types and knowledge domains (e.g., algebraic, geometric) to filter candidate nodes. This introduces a degree of human supervision and limits the method’s autonomy and scalability. Especially if we concider going beyond mathematical domain.
>
> As stated in the paper (Sec. 4.3, line 282), we used manual tags simply for clarity in the initial implementation. **But these tags can be, and in practice are, easily generated automatically by lightweight encoder-only models.**  Further, we only used knowledge tags at the inference, while template type was additional metadata that had. A rich literature shows that small models achieve high accuracy in such classification tasks [1, 2]. For the new experiments on the GPQA dataset as well we used tagging. This demonstrates that manual tagging is not required and can be easily automated.
>
> Beyond mathematics, many domains have clear top-level taxonomies (e.g., scientific disciplines, programming tasks, law/finance categories), and tagging a query or reasoning step into these domains is a standard and well-studied in information retrieval [3].
> We will update the paper to reflect both improvements: the new GPQA experiments, which show that RoT scales beyond mathematical reasoning, and the automated tagging procedure.
>
> _We hope these additions address the reviewer's concerns. If so, we would greatly appreciate a corresponding score adjustment. Otherwise, we are happy to continue the discussion._
>
> **References**
>
> [1] Piotr Piekos, Henryk Michalewski, and Mateusz Malinowski. Measuring and improving Bert’s mathematical abilities by predicting the order of reasoning. ACL 2021.
>
> [2] Shuai Peng, Ke Yuan, Liangcai Gao, and Zhi Tang. Mathbert: A pre-trained model for mathematical formula understanding.
>
> [3] Daria Alexander, Wojciech Kusa, Arjen P. de Vries. ORCAS-I: Queries Annotated with Intent using Weak Supervision. SIGIR 2022.
>
> [4] HiExpan: Task-Guided Taxonomy Construction by Hierarchical Tree Expansion. Jiaming Shen, Zeqiu Wu, Dongming Lei, Chao Zhang, Xiang Ren, Michelle T. Vanni, Brian M. Sadler, Jiawei Han. SIGKDD 2018.
>
> [5] Rein, David, Betty Li Hou, Asa Cooper Stickland, Jackson Petty, Richard Yuanzhe Pang, Julien Dirani, Julian Michael, and Samuel R. Bowman. Gpqa: A graduate-level google-proof q&a benchmark. COLM 2024.

---

> > ### Author Response · Authors · 2025-11-26
> > **Kind reminder to reviewer**
> >
> > This is a kind reminder that we have provided updated responses and two additional experiments addressing your comments. We would appreciate your review.

---

### Author Response · Authors · 2025-12-02
**Summary of Rebuttal and Reviewer Feedback**

Two reviewers explicitly indicated that the rebuttal fully addressed their concerns. Reviewer **ZE4Z** now *strongly recommends accepting* the paper and emphasizes that reusing thoughts is an important and general mechanism for efficient LLM reasoning. Reviewer **VNeJ**, who initially gave a lower score, stated that they would like to increase their score in light of the rebuttal but are unable to do so due to the recent ICLR OpenReview policy changes (score updates being disabled). We are very grateful for these positive updates and for all three reviewers’ thoughtful and detailed feedback.

Reviewers **UvKG**, **ZE4Z**, and **VNeJ** highlighted several strengths of our work: RoT’s **novel dynamic retrieval-based reasoning framework and thought graph**, its ability to **assemble context-specific templates on the fly via reward-guided traversal** (UvKG), its connection to **compositional memory and human-like “connecting the dots” analogical reasoning** (UvKG, ZE4Z), its **flexible reuse of granular reasoning steps beyond fixed scaffolds** (ZE4Z), and its **conceptually elegant, technically sophisticated, and methodologically rigorous design** with formal definitions, curated templates, and comprehensive evaluations and ablations (ZE4Z, VNeJ). RoT was recognized as directly tackling the **efficiency–accuracy tradeoff** for practical, cost- and latency-aware LLM deployment (ZE4Z).
Below, we summarize the main common concerns raised by the reviewers and how our new experiments and clarifications address them:

> **Concern C1 (UvKG, ZE4Z, VNeJ): Generality beyond mathematics.**

We extended RoT to the **GPQA, a scientific-reasoning benchmark**, constructing a scientific thought graph and applying the same RoT pipeline. Across Qwen3-0.6B–14B, **RoT/RoT+TI match or improve CoT accuracy while substantially reducing output tokens**, showing that RoT generalizes to non-math, less formulaic reasoning tasks.

> **Concern C2 (VNeJ): Generality across model families and broader evaluation.**

Beyond the original Qwen3 family, we now evaluate **DLER-7B** and **DeepSeek-R1-Llama8B** on AMC 2023 and AIME 2024. In all cases, **RoT/RoT+TI maintain or improve accuracy and often reduce tokens**, yielding new Pareto-efficient points in the accuracy–efficiency trade-off and demonstrating that RoT is not tied to a single model family.

> **Concern C3 (UvKG, ZE4Z, VNeJ): Robustness to near-duplicates, tagging, and parameter stability.**

We designed a **near-duplicate experiment** by creating structurally identical AIME/AMC variants with different numerical values, building a thought graph from these variants, and evaluating RoT on the original problems. RoT/RoT+TI remain robust and often improve accuracy with fewer tokens, confirming that RoT retrieves **abstract reasoning steps** rather than copying conflicting numeric constants. We also clarify that template/domain tags are now **automatically generated by lightweight encoder models** (as in our GPQA experiments) and that the semantic parameters $\alpha$ and $\tau$ are **kept fixed across math and scientific settings**, yet perform robustly, indicating they are not overfitted to a single dataset.

We will incorporate these new experiments and clarifications into the revised version and hope that this strengthened evidence fully addresses the main concerns.

---

### Meta-Review · Area_Chair_ifPe · 2025-12-29

**Summary:**

This submission proposes RoT, a thought-reuse framework for LRM efficient reasoning. The authors addressed most of the reviewers’ concerns via new experiments. Reviewer ZE4Z strongly recommends acceptance; Reviewer VNeJ will raise the score. The work is novel and robust, meeting acceptance criteria.

**Reviewer Concerns:**

Resolved concerns: generality beyond math (GPQA), cross-model family adaptability, robustness to near-duplicates. All addressed with new experiments and clarifications.

No unresolved critical issues.

**Reviewer Scores:**

Reviewer VNeJ (initial 2) will raise the score as the concerns are all addressed.

---

### Decision · Program_Chairs · 2026-01-26

Accept (Poster)